# Nanotechnology Applied to Cellulosic Materials

**DOI:** 10.3390/ma16083104

**Published:** 2023-04-14

**Authors:** Ana Fernandes, Luísa Cruz-Lopes, Bruno Esteves, Dmitry Evtuguin

**Affiliations:** 1Campus Universitário de Santiago, University of Aveiro, 3810-193 Aveiro, Portugal; ana.augusta@ua.pt; 2Department of Environmental Engineering, Polytechnic Institute of Viseu, Av. Cor. José Maria Vale de Andrade, 3504-510 Viseu, Portugal; lvalente@estgv.ipv.pt; 3Centre for Natural Resources, Environment and Society-CERNAS-IPV Research Centre, Av. Cor. José Maria Vale de Andrade, 3504-510 Viseu, Portugal; 4Department of Wood Engineering, Polytechnic Institute of Viseu, Av. Cor. José Maria Vale de Andrade, 3504-510 Viseu, Portugal; 5CICECO—Aveiro Institute of Materials and Department of Chemistry, University of Aveiro, 3810-193 Aveiro, Portugal; dmitrye@ua.pt

**Keywords:** cellulose nanocrystals, cellulose nanofibers, bacterial cellulose, nanogenerators

## Abstract

In recent years, nanocellulosic materials have attracted special attention because of their performance in different advanced applications, biodegradability, availability, and biocompatibility. Nanocellulosic materials can assume three distinct morphologies, including cellulose nanocrystals (CNC), cellulose nanofibers (CNF), and bacterial cellulose (BC). This review consists of two main parts related to obtaining and applying nanocelluloses in advanced materials. In the first part, the mechanical, chemical, and enzymatic treatments necessary for the production of nanocelluloses are discussed. Among chemical pretreatments, the most common approaches are described, such as acid- and alkali-catalyzed organosolvation, 2,2,6,6-tetramethylpiperidine-1-oxyl (TEMPO)-mediated oxidation, ammonium persulfate (APS) and sodium persulfate (SPS) oxidative treatments, ozone, extraction with ionic liquids, and acid hydrolysis. As for mechanical/physical treatments, methods reviewed include refining, high-pressure homogenization, microfluidization, grinding, cryogenic crushing, steam blasting, ultrasound, extrusion, aqueous counter collision, and electrospinning. The application of nanocellulose focused, in particular, on triboelectric nanogenerators (TENGs) with CNC, CNF, and BC. With the development of TENGs, an unparalleled revolution is expected; there will be self-powered sensors, wearable and implantable electronic components, and a series of other innovative applications. In the future new era of TENGs, nanocellulose will certainly be a promising material in their constitution.

## 1. Introduction

Cellulose was discovered by Payen Anselme, a French chemist, in 1838 [1] It is a renewable biopolymer, the most abundant in nature, with estimated natural production of about 1.5 × 10^12^ tons per year [2]. Besides being abundant, cellulose is nontoxic, biocompatible, and biodegradable. The main sources of cellulose are plants, algae, bacteria, and tunicates (marine invertebrates) [3,4]. Cellulose is a natural macromolecule formed by 300–15,000 units of β-d-glucopyranose, bound by β-1,4-glycoglycodic linkages, and possessing an amorphous–crystalline supramolecular structure [5] Cellulose’s crystalline counterpart is rod-shaped and is found in the structure of elementary fibrils (EFs), where crystalline and amorphous regions alternate [5]. EFs are assembled in microfibrils and macrofibrils, which in turn make up the cell skeleton in plants (Figure 1). Cellulose crystallites can be isolated from native celluloses in different morphologic forms, and these so-called nanocelluloses are nanomaterials. By definition, nanocelluloses have at least one dimension in the nanometric range; i.e., they have a dimension of less than 100 nm [6,7], according to ISO/TS 20477:2017 [8,9]. Due to their unique properties, nanocelluloses can be used in numerous applications in different advanced materials, one of the most exciting being electronic devices and triboelectric nanogenerators (TENGs).

This review focuses on the methods for obtaining nanocellulose and its nanotechnological applications, and is composed of two main parts. In the first part of the review, a brief characterization of the three cellulose nanomorphologies is performed, namely cellulose nanocrystals (CNCs), cellulose nanofibers (CNFs), and bacterial cellulose (BC), along with their preparation methods. It should be noted that the terms cellulose nanocrystals, cellulose nanofibers, and bacterial cellulose are strictly established in ISO/TS 20477: 2017 [8] The methods of nanocellulose preparation include such approaches as “*bottom-up*” or “*top-down*” [7,10] The mechanical, chemical, and enzymatic methods belong to a “*top-down*” approach for the production of cellulose nanocrystals (CNCs) and cellulose nanofibers (CNFs), while the “*bottom-up*” approach is commonly used for BC production. In relation to mechanical methods, methods for preparing CNFs through the disaggregation of cellulose fibers via defibrillation are reviewed. Within the category of mechanical methods, the following approaches are considered: refining, high-pressure homogenization, microfluidization, grinding, ball-milling, cryogenic crushing, steam explosion, ultrasound, extrusion, aqueous counter collision, and electrospinning [11,12]. For these mechanical methods, their functioning is presented, along with some illustrative schemes. In the scope of mechanical methods, there was also the aim to gather a set of investigations whose objective was to evaluate how operational parameters influenced the properties of CNFs. Thus, for each mechanical method and its operational parameters that interfere with the properties of CNF, a set of investigations was gathered in this study whose objective was to evaluate the relationship between the operational parameters and the properties of CNFs.

The chemical pre-treatment methods commonly applied before mechanical treatments were also reviewed. These are mainly intended for the production of CNFs. Among chemical pretreatments, acid- and alkali-catalyzed organosolvation, TEMPO-mediated oxidation, oxidation with APS and with SPS, ozone pre-treatment, extraction with ionic liquids [12,13,14,15], and two innovative methods for obtaining CNFs were described. One of these new methods consists of simultaneous bleaching and oxidative modification with TEMPO [16] and the other employs sodium persulfate (SPS) with ultraviolet light [17]. This review also presents the chemical treatments for the production of CNC, which are essentially variations of acid hydrolysis, as it degrades the amorphous part of the cellulose, leaving the crystalline part in the form of CNC [12,15]. Regarding chemical methods of CNC production, it should be noted that the TEMPO system, which has been used for the preparation of CNFs (the so-called TEMPO-CNF), is also suitable for CNC production [18]. This review also covers enzymatic hydrolysis pretreatment, considered by some researchers as the “most ecological route” for CNF production [13,15]..Enzymatic hydrolysis is carried out by cellulases, which more rapidly degrade the disordered cellulose counterpart of fibers. The three types of cellulases (endocellulases, exocellulases, and β-glucosidases), as well as their degradation modes, are specified. The first part of the review also contains a set of approaches in which chemical and enzymatic treatments were successfully combined in the production of nanocelluloses (CNFs and CNCs), and the properties of these nanocelluloses (diameter, length, and crystallinity index) are discussed.

In the second part of this review, recent developments in the innovative applications of nanocelluloses are summarized.

These examples clearly demonstrate the enormous potential of nanocellulose applications in different areas, from food, health, cosmetics, pharmaceuticals, medicine, and water treatment to electronic devices. The functioning of TENGs is also reviewed for the first time and examples of these devices produced from BC, CNCs, and CNFs are discussed.

## 2. Nanocellulose Nanomorphologies

Cellulosic materials can be converted into cellulose nanofibers (CNFs) and nanocrystals (CNCs), depending on the mechanical and chemical treatment applied (Table 1). In addition to these two nanomorphologies, bacterial nanocellulose, BC, also exists. These three types of nanocellulose (CNF, CNC, and BC) have different average sizes (Table 1). However, it should be noted that, for each type of nanocellulose, the values of the average sizes found in the literature present slight differences. These differences are mainly due to the different sources used and the various methods of preparation [12,19].

It should be noted that, in the literature, the terminology used for the three types of nanocellulose is not concordant, so there are several designations for the same type of nanocellulose. For cellulose nanocrystals (CNCs), the following terms are used as synonyms: nanocrystalline cellulose, rod-type cellulose microcrystals, nanocellulose whiskers, and nanowhiskers [4,20,21]. Regarding cellulose nanofibers (CNF), the synonyms used are cellulose nanofibrils, nanofibrillated cellulose, or nanocellulose [1,4,20,22]. Additionally, bacterial cellulose (BC) is also called microbial cellulose [1,13,20,22].

Cellulose nanofibers (CNFs) are isolated from cellulose through simple mechanical disaggregation before and/or after chemical (or enzymatic) treatments [1,22]. CNFs were first produced in 1983 [12,22], and the first method of manufacture of CNF was patented in 1985 [12]. CNFs are composed of long, flexible, and scrambled “spaghetti-type” wires, with diameters below 100 nm and lengths of several micrometers [22]. In these preparations, CNFs contain amorphous and crystalline regions [23], as designated in Figure 2.

Cellulose nanocrystals (CNCs) appear “rice-like”, and are shorter and less flexible than CNFs (Figure 2). The reason for their lower flexibility is due to the preparation method, because they are obtained by acid hydrolysis (or enzymatic), which causes the removal of amorphous regions of cellulose; thus, they are less flexible than CNF [4,22]. The first CNCs obtained by acid hydrolysis were produced by Nickerson and Habrle in 1947 [12,15]. It should be noted that cellulose nanofibers (CNFs) and nanocrystals (CNCs) are obtained by “top-down” approaches.

Bacterial nanocellulose, BC, was first reported in 1886 by A.J. Brown [9]. BC is produced from glucose by bacterial families through a “bottom-up” approach. These bacteria, such as Gluconacetobacter xylinus, are grown in aqueous media and BC is produced by synthesizing cellulose and nanofibers in a process that takes a few days [22]. There are several families of bacteria that produce BC, namely Achromobacter, Alcaligenes, Aerobacter, Agrobacterium, Azotobacter, Komagataeibacter, Pseudomonas, Rhizobium, Sarcina, Dickeya, and Rhodobacter [24]. The most studied bacteria for this purpose are Komagataeibacter, (formerly Gluconacetobacter) [24]. BC does not contain lignin or hemicellulose, and it is said to have no impurities [19], having a high degree of purity when compared with nanocrystals and cellulose nanofibers [22]. BC is synthesized extracellularly, so hydrogen bonds between fibrils are more intense when compared with those in plant cellulose [1]. For these reasons, BC has properties that make it very attractive for various applications. These properties are its high degree of crystallinity, water retention capacity. and tensile strength [2]. The dimensions of bacterial cellulose vary depending on the family of bacteria, the conditions of cultivation, and the type of bioreactor [14].

ISO/TS 20477: 2017 [8] defines CNFs as network structures with dimensions from 3 to 100 nm, lengths that can reach 100 μm, and an aspect ratio of >10 (aspect ratio is the quotient between length and diameter). For CNCs, this ISO standard establishes a diameter of 3 to 50 nm and length from 100 nm to several microns (μm), with the aspect ratio usually in the range of 5 to 50.

## 3. Nanocellulose Preparation Methods

For the preparation of nanocelluloses, various types of treatment methods (physical, chemical, and enzymatic) are used. Below, some methods for the various types of treatments are listed.

### 3.1. Physical Methods

Mechanical methods for the defibrillation of wood pulp before and/or after chemical (or enzymatic) processes are used to prepare cellulose nanofibers. There are several mechanical methods that can be applied to produce CNFs. These methods, although different from each other, have the same objective, which is to break the hydrogen bonds and the van der Waals forces between the cellulose layers to cause the individualization of the microfibers of cellulose in CNFs. There are several mechanical methods, namely refining, high-pressure homogenization, microfluidization, grinding, ball-milling, cryogenic crushing, steam explosion, ultrasound, extrusion, aqueous counter collision (ACC), and electrospinning [11,12]. This review describes each of these physical methods. It should be emphasized that, in addition to the description of each physical method, works whose objective was to evaluate how the operational parameters influence the properties of CNFs were also included.

Refining is a physical method in which the pulp is moved through a hole between two discs (one that rotates and one that is fixed) [12]. The surfaces of these discs have grooves and bars, against which the pulp undergoes cyclic and repeated stresses [1,25]. A study on how energy and the number of cycles performed with an ultra-disc refiner affect the physical–structural, morphological, and thermal properties of CNFs obtained from bleached cellulose of kraft eucalyptus is presented [26]. In this study, a two-disc ultra-refiner was used (one with rotational movement at 1600 rpm and the other being stationary). The researchers came to several conclusions, one of which was that a higher degree of defibrillation led to the production of CNFs with a smaller diameter and more homogeneous suspensions in terms of diameter, but a considerable increase in energy was required to only slightly increase transparency and viscosity. They also concluded that there was an optimal defibrillation degree value, beyond which the crystallinity of the CNFs may have been lost [26]. There have been studies that indicate that 16 to 30 passes in a refiner increase the mechanical strength of the fibers [1].

High-pressure homogenization is another physical method in which cellulose pulp is required to pass through a pressurized valve [12] to 50–2000 MPa [13,27] (Figure 3A).

It is a disruptive method that causes shear and turbulence forces, and, consequently, the rupture of the cellulosic structure. The first time this method was applied to CNF preparation was in 1983 [25]. Studies show that the degree of defibrillation depends on the number of cycles, as well as on the applied pressure [13]. In a review, three drawbacks of homogenization were pointed out, including the frequent occurrence of clogging, the high energy consumption, and the damage to the CNF structure [13]. To reduce these inconveniences, some proposals were made, namely including another mechanical method before homogenization (e.g., refining or other) to prevent clogging or the application of chemical pre-treatment to decrease energy consumption [13]. A homogenizer can consume up to 70 MW h/t; this value, however, is drastically reduced to 2 MW h/t if a chemical pre-treatment is applied [15]. The effects of refining and high-pressure homogenization, the development of CNFs, and energy consumption were studied [28] using bleached eucalyptus kraft pulp. The researchers concluded that both methods (refining and homogenization) decreased the diameter of CNF and increased the aspect ratio (length/diameter ratio), but homogenization was more effective in changing these two parameters. A surprising result was that the more refined sample showed a residual increase in resistance compared with the less refined sample, but had ten-fold higher energy consumption [28].

Microfluidization is a process very similar to the process of high-pressure homogenization [12,25]. In the microfluidization process, the cellulose pulp enters a pressurized valve through Z-shaped channels [12] (Figure 3B). However, unlike a homogenizer operating at constant pressure, a microfluidizer works at a constant shear rate [13]. To achieve the desired degree of fibrillation, it is necessary to repeat the number of passes in the microfluidizer and use cameras of different sizes [13]. Studying how these two parameters affect the properties and morphology of CNFs was the objective of Taheri and Samyn, (2016). They concluded that there was an optimal number of passages and camera size, because there is a compromise between the amount of fibrillation and the stability of the suspension. It was found that a smaller interaction chamber resulted in lower crystallinity, while a greater number of passages increased the crystallinity of the fiber, although this increase was not significant [29]. One of the concerns was the energy consumption required for the production of CNFs. Studies have revealed that the lower the value of the energy used, the lower the fibrillation and the lower the amount of CNF produced [30].

The grinding method consists of using two grinding stones, one fixed and one moving, at about 1500 rpm [12,13]. Cellulose pulp passes between the two stones (Figure 3C). The degree of fibrillation depends on the distance between the stones, the shape of the stones, and also the number of passages [13]. In a review article, a comparison between grinding and high-pressure homogenization was made [15]. The review highlighted the advantages of the grinding method, including reduced energy consumption, a lower tendency for clogging, and high efficiency. However, they also highlighted grinding’s disadvantages, including low crystallinity and the low thermal stability of CNFs [31] In a work that used pine cones (Jack pine: Pinus banksiana Lamb) that were free of extractives for CNF production, after alkaline pre-treatment, grinding was performed and optimized [31] It has been reported that, when the number of cycles increases, the tensile strength and the modulus of elasticity of the fibers increase as well, but continuing to increase the number of cycles causes these properties to decrease, thus indicating that there is an optimal value.

There have been at least two recent reviews on ball-grinding [23,32]. One of these reviews classifies this method as a green technology for the preparation of nanocellulose [32]. Ball-milling for pulp extraction was first performed in 1940 [23]. A ball mill, as the name suggests, is formed by a hollow cylinder filled with balls; the balls are usually steel, rubber, ceramic [32], or zirconia [12]. The operation of a ball mill is simple; the mill has rotational movement and, when rotating, generates shear and friction forces between the balls and between the balls and the surface of the mill, which break the cellulose fibers [23]. In one of the above reviews, several advantages of grinding balls have been highlighted, namely the simplicity of operation, low cost, speed, and reproducible results due to the ease of speed control. However, some disadvantages have also been highlighted, such as noise generation, long operating times, high contamination, and the formation of irregular fibers [32]. As for the types of mills, there are three categories: ball mills, vibrators, and planetariums [23,32] (Figure 3D). A ball mill is a hollow cylinder with balls that rotate around the shaft. In this type of mill, one of the parameters that influences the effectiveness of the process is the diameter of the mill, as the diameter is the fall height of the balls. The larger the diameter, the greater the energy of the balls and, consequently, the greater the degree of fibrillation [32]. Vibratory mills are named as such because they have vibration movements, forward and backward, with high vibrational frequencies. In this type of mill, the degree of fibrillation depends on the characteristics of the vibration, i.e., the frequency and amplitude of the vibration [32]. Planetary mills are formed by several containers that are placed on a disc that rotates [32]. Each container simultaneously performs a rotational motion around its axis and a translational motion around the disc axis. Regardless of the type of ball mill, there are several parameters to be taken into account in the operational conditions and that influence the properties of CNF, namely the speed of rotation, the grinding time, the number of balls, the size of the balls, the dry or moist state of the material to be ground, and the relationship between the weight of the balls and the material to be ground [23]. There have been several studies on the influence of these parameters on CNF properties. The influence of the grinding time on the properties of microcrystalline cellulose (MCC) has been studied [33]. Although this study was on the micro-scale and not on the nano-scale, it seems pertinent to report its results here. In this study, a planetary mill with zirconium dioxide balls (6–10 mm) was used. The influence of grinding time on the morphological structure, crystallinity index, and thermal stability of MCC powders was analyzed. There were several consequences of the increase in grinding time, namely a decrease in particle size, destruction of crystalline areas, decreased crystallinity, change in shape and stem to a spherical shape, and no change in molecular structure. It was also verified that the MCC powders prepared with a shorter grinding time exhibited better thermal stability, and with a longer grinding time, the viscosity decreased [33]. Another study aimed to investigate the effect of several parameters (grinding time, ball size, ball mass ratio: cellulose, and alkaline concentration) on CNF production by ball-grinding [34] These parameters were evaluated and combined. Given the high number of variables, multi-factor analysis (MFA) by the Taguchi method and ANOVA analysis were used. The researchers concluded that the size of the balls was the parameter that could be disregarded, and that the mass ratio of the balls and cellulose was the most determining parameter. L. Zhang et al. (2015) conducted a study of the following parameters in CNF production: proportion of ball and cellulose mass, grinding time, ball size, and alkaline pre-treatment [35] They concluded that the grinding time and the proportion of ball mass were the two parameters determining the morphology of CNFs and that alkaline pre-treatment helped to weaken the hydrogen bonds between cellulose fibers, but warned that pre-treatment could damage CNFs.

The cryogenic crushing method consists of freezing the cellulose pulp in liquid nitrogen to freeze its contents in water. Then, the frozen fibers are crushed with a grinder; this procedure breaks the cell walls due to the pressure of the ice [12,15]. This method has high energy consumption and low productivity, so it is a physical method that is quite expensive [13]. Several articles used this method, usually combined with other mechanical methods of CNF production, one of which used refining followed by cryogenic crushing [36] However, no study was found on the influence of the possible operational parameters of cryogenic crushing on the properties of CNFs.

Steam explosion is a thermomechanical method performed in an autoclave within a small time interval, but at a high temperature (180–210 °C). In this method, the heat transported by steam penetrates, by diffusion, the cellulose fibers; then, an explosive decompression is made. This pressure discharge suddenly promotes the rupture of glycolipid bonds and hydrogen bonds, and, as a consequence, the separation of fibers [12]. This method was first used in 1995 to transform wood into fibers [1]. A study produced CNFs from Japanese cedar (*Cryptomeria japonica*) using the steam explosion method, followed by extraction with water and acetone [37]. In this work, the influence of this method on the morphological, chemical, and mechanical properties of CNFs was evaluated, and it was concluded that an increase in the severity of vapor explosion drastically decreased the molecular weight of α-cellulose in the exploded pulp. Steam explosion is considered an environmentally friendly method, but it has some drawbacks. Experts have pointed out that this method gives rise to cellulose nanofibers with different sizes and low quality; to overcome this inconvenience, they suggest a subsequent chemical treatment (with hydrogen peroxide), as well as ultrasound treatment [27].

For ultrasounds to be used as a physical method for the formation of cellulose nanofibers (CNFs), it is necessary to have a liquid medium for the cavitation phenomenon to occur. This phenomenon occurs because an ultrasound is a mechanical wave. Like any sound, an ultrasound is formed by rarefaction zones and compression zones alternating with each other. Thus, when an ultrasound propagates in a liquid, it forces the liquid particles to separate and form bubbles that grow, and when they reach a critical size, violent collapse occurs [38]. Ultrasounds lead to the formation of CNFs when applied to cellulose pulp in an aqueous medium, because cavitation bubbles are formed that burst and lead to the formation of CNFs [12]. This is because the cavitation phenomenon leads to temperature and pressure values higher than 5000 °C and 500 atm, respectively, which are conditions that promote the effective fibrillation of cellulose in nanofibers [14].

Ultrasounds are also common for improving the dispersion of cellulose nanocrystals (CNCs) in various polymeric solvents and matrices [39] In one such study, CNC examples obtained by hydrolysis were treated with ultrasound [40]. It was concluded that ultrasounds affected the disintegration of CNCs and, if the time of action of the ultrasounds was high, degradation of the same occurred.

Recently, Shojaeiarani et al. evaluated the effect of the amplitude and time of ultrasounds on the morphology of cellulose nanocrystals and their dispersion [39]. They concluded that the increase in amplitude caused a marked decrease in the length of the CNC, while the increase in the time interval of the ultrasounds slightly affected the length of the CNC [39].

In another study, the influence of hybrid treatment via ultrasounds combined with chemical pre-treatment on the dimensions and appearance of CNFs was evaluated [7]. The authors concluded that applying chemical pre-treatment only resulted in the agglomeration of the CNFs, but by combining this pre-treatment with ultrasound, this inconvenience could be avoided.

During the extrusion process, the rupture of cellulose fibers occurs when the cellulose pulp is subjected to double screw extrusion. Operational parameters, such as the screw speed, cylinder temperature, and humidity, influence the properties of CNF [12]. This method was successfully used for CNF production, without pre-treatment [41]. According to these researchers, the method has two advantages over other methods, such as homogenization or grinding. One advantage is that this method allows the treatment of samples with a relatively high solids content; the other advantage is the production of a nanocellulose powder (instead of a paste or aqueous solution), which is advantageous for transport and storage. The effect of the number of passages through an extruder with a double screw on the properties of CNFs was evaluated. The two properties evaluated were the degree of degradation and the viscosity of the fibers. They concluded that fibrillation was only possible until a certain number of passages through the extruder; after this optimal value, the CNFs degraded. A review of nanocellulose production methods indicated that double-screw extrusion allowed samples with a high solids content to be obtained and indicated that this content could be between 25% and 40% by weight [15] Recently, a study was published in which extrusion was applied (with a double screw) alongside enzymatic hydrolysis in situ [42]. The researchers concluded that bio-extrusion decreased the tendency of cellulose nanofibers to aggregate.

The aqueous collision counter-collision method (ACC), as the name suggests, consists of using repeated cycles with two jets of aqueous cellulose suspensions, one against the other, at high pressure [12,15] (Figure 3E). The collision energy of the jets induces the breakdown of hydrogen bonds and van der Waals forces in cellulose fibers and, consequently, the release of CNFs [43,44]. In this ACC method, the operational parameters of the jet pressure and cycle time, influence the size and morphology of nanocellulose [45]. This method was also used for CNF production from wood and bacterial cellulose [46] and it was possible, through this method, to control the width of the CNFs by adjusting the speed of the jets properly. In that study, the researchers, using the water ejection pressure, calculated the kinetic energy of the water molecules and compared the value obtained with the energies of the hydrogen bonds and the van der Waals forces, and concluded that certain jet pressures had enough energy to break the hydrogen bonds. The ACC method is widely used to produce bacterial nanocellulose (BC) because it transforms natural cellulose in the crystalline phase (metastable, Iα) into stable cellulose (Iβ) without decreasing the degree of crystallinity [43,46]. Studies show that BC films treated with ACC decrease in diameter with an increase in the number of cycles [43,44].

The electrospinning method consists of placing a cellulose solution to drip and applying an electric field to the drops. This method is considered economical [1] and is one of the most popular methods, not only for preparing CNFs, but also for preparing other nanofibers [11]. This method was patented in 1934 by Formhals A. [47], as stated before [1]. In a recent review on electrospinning, it was stated that a device for performing this method generally has four parts, that is a syringe with the polymer solution, a metal needle, a power supply, and a metal collector [48]. Owing to the application of an electric field to the drops, a cone is formed, the so-called Taylor cone (Figure 3F). According to this review, several parameters affect the nanofibers obtained by this method, including the electrospinning parameters (applied electric field, distance between needle and collector, needle diameter, and flow of solution through the needle), solution parameters (solvent, polymer concentration, viscosity, and conductivity of the solution), and environmental parameters (temperature and humidity) [48]. Another review focused on electrospinning cellulose using ionic liquids, and on the processes and applications of this method [21]. In the review, it was emphasized that electrospun nanocellulose with ionic liquids had applications in various areas, from water treatment to health and energy storage, among others. It should be noted that cellulose is insoluble in most organic solvents; thus, a large number of publications on electrospun CNFs used cellulose acetate as a soluble precursor. However. there are reports of the use of pure CNF [11]. Other solvents currently used in the electrospinning of cellulose fibers, in addition to ionic liquids, are ethylenediamine salts [1]. Table 2 shows the above-mentioned studies. There have also been some studies that evaluated different mechanical methods, including homogenization, ultrasound, and grinding, and their effects on the properties of nanocrystals (CNCs) [49]. It was concluded, in that work, that the CNCs produced by homogenization exhibited lower thermal stability, despite having a good crystallinity index (74.7%), and smaller sizes than those obtained with ultrasound and grinding. Another study aimed to evaluate microfluidization with grinding as a method for CNC preparation [50]. The study concluded that microfluidization had the advantage of being faster and more effective than grinding.

**Figure 3 materials-16-03104-f003:**
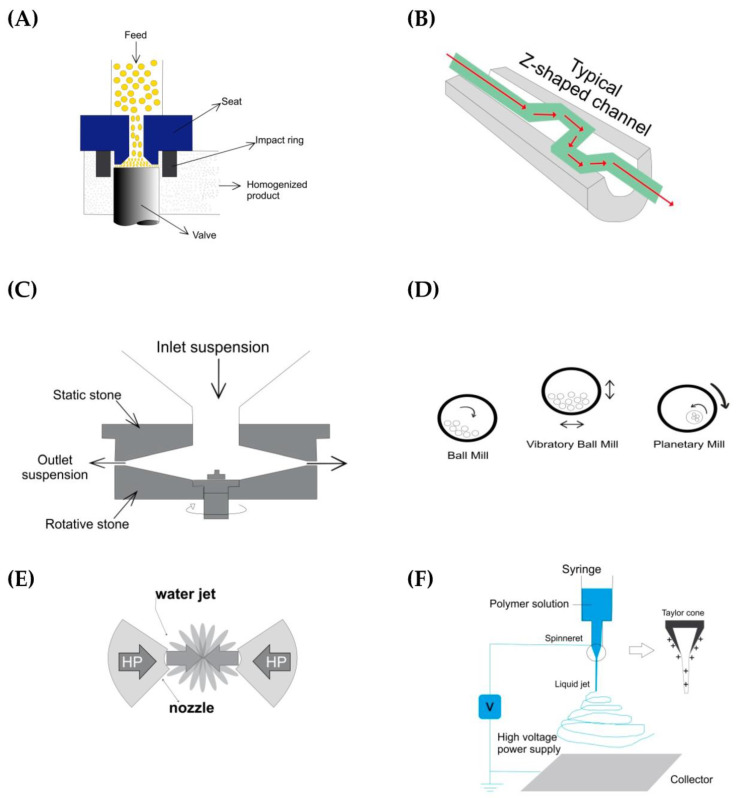
Some mechanical methods: (**A**) High pressure homogenizer [51]; (**B**) Microfluidizer [52]; (**C**) Grinding [53]; (**D**) Ball mills [32]; (**E**) Aqueous Counter Collision (ACC) [44] (**F**) Electrospinning [54] (The schemes have been adapted).

### 3.2. Chemical Methods

Acid hydrolysis is one of the most common chemical pretreatment methods for producing nanocellulose from purified cellulosic materials [23] By applying acid hydrolysis, the amorphic parts of the cellulose are degraded, leaving the crystalline parts, which, due to the compact structure, do not allow the penetration of the acid into crystallites [15]. Therefore, after acid hydrolysis, cellulose nanocrystals (CNCs) remain [15,23]. Acid hydrolysis was performed for CNC production for the first time in 1947, by Nickerson, R.F. & Habrle, J.A. [55]. For hydrolysis, the most commonly used acid is sulfuric acid [56]. In hydrolysis with sulfuric acid, sulfate ions react with the hydroxyl groups of cellulose, forming a semi-ester (${OSO}_{3}^{-}$) due to the negative charge deposited on the surfaces of the CNCs, generating repulsive forces that prevent the aggregation of nanocrystals [23,56,57]. In a review, it was stated that, for acid hydrolysis with sulfuric acid, the concentration of acid should be in the range of 60–65%, the temperature should be 40–50 °C, and the hydrolysis time should be 30–60 min [15].

Regarding acid hydrolysis, a recent work in which CNCs were produced from ramie fibers (*Boehmeria nivea*) is highlighted [58]. In that study, after chemical purification, the optimization of acid hydrolysis was performed with sulfuric acid (58%), and the temperature of 45 °C and the time of 30 min were found as ideal conditions. Under these conditions, cellulose nanocrystals (CNCs) with a diameter of 6.67 nm, length of 145.61 nm, and high crystallinity index (90.77%) were produced. This work confirmed that, in acid hydrolysis, it is necessary to control the parameters, such as temperature and reaction time, in addition to the type and concentration of acid. In addition to these parameters, the properties of CNCs also depend on pre-treatment and post-treatment, as well as on the source of cellulose used [56]. A recent study [59] proposed the integration of the environmentally friendly heterogeneous catalytic processes of wood hemicellulose hydrolysis and peroxide delignification for the biorefinery of birch wood into microcrystalline, microfibrillated, and nanocrystalline celluloses, xylose, and sorbents. This process yielded 41.2% microcrystalline cellulose that, after sulfuric acid hydrolysis and ultrasonic treatment, produced microfibrillated cellulose with a yield of 31.8% and nanocrystalline celluloses with a yield of 10.9%. A similar approach was used to obtain microfibrillated cellulose with a low degree of polymerization from spruce wood pulp, which made it possible to decrease the number of stages, decrease energy costs, and increase the range of applications, according to the authors [60].

The main disadvantage of acid hydrolysis is the need for pH control and washing with water. This washing of the cellulose suspension is carried out until a neutral pH is obtained. An alternative is to add an alkaline solution to neutralize this suspension [23]. In addition to this disadvantage, acid hydrolysis has other drawbacks, such as the corrosion of the equipment due to the application of strong acids, and the environmental problem of the disposal of the residual acid [14,15,61].

Recently, in a study, CNCs were produced from cotton using acid hydrolysis assisted by ultrasound [62]. The researchers compared the results obtained with other published results for acid hydrolysis without ultrasound for the same source of cellulose (cotton) and concluded that acid hydrolysis with ultrasound was a promising technique for CNC production, as it allowed CNCs with smaller sizes and a higher crystallinity index to be obtained (Table 3).

Pre-treatments are usually used before defibrillation with the aim of removing lignin and hemicellulose, and making the cellulose structure weaker and thus more conducive to transformation into nanofibers [19]. During the production of CNFs, the main cost is the energy consumed in mechanical methods [15,63]. Thus, chemical pre-treatments are performed before mechanical treatments, also with the purpose of reducing the energy of mechanical treatments, by reducing the number of passages through the various defibrillation devices, in addition to preventing the clogging of this machinery [15]. The review explained how chemical pre-treatments are applied for CNF production, including solvent-assisted pre-treatment, organic acid hydrolysis pre-treatment, oxidation moderated by TEMPO (2,2,6,6-tetramethylpiperidine-1-oxyl), periodate–chlorite oxidation, oxidative sulfonation, carboxymethylation, cationization, and extraction with ionic liquids or deep eutectic solvents [15]. The same review mentioned chemical pre-treatments for CNC production (in addition to acid hydrolysis and enzymatic hydrolysis), oxidation degradation (via TEMPO or via APS (ammonium persulfate)), and extraction with ionic liquids. In another, more current review, the authors mentioned, for the production of CNF, the following chemical pre-treatments: alkaline, acid, oxidation (with hydrogen peroxide, oxygen, and ozone), extraction with ionic liquids, and extraction with other solvents (alcohols, ethers, ketones, and benzene) [14] Similarly, Yadav et al. suggested, as chemical pre-treatments for CNF preparation, alkaline pre-treatment, enzymatic pre-treatment, oxidation moderated by TEMPO, carboxymethylation, and acetylation, and for CNC production, only acid hydrolysis. The review aimed to present some of the above-mentioned chemical pre-treatments. Two recently published innovative methods were also described, one using bleaching and TEMPO at the same time [16], and another that employed sodium persulfate (SPS) and ultraviolet light [17].

An alkaline pre-treatment facilitates the breaking of links between lignin and cellulose [12] by breaking the OH link through the introduction of ions into the fibers, which contributes to the extraction process. The process can be translated by the following scheme [12]:Fiber-OH + NaOH→ Fiber-O-Na^+^ + H_2_O(1)

Alkaline pre-treatment can be applied not only with sodium hydroxide (NaOH), but also with other alkaline solutions, such as potassium hydroxide (KOH), calcium hydroxide (Ca (OH)_2_), ammonia (NH_3_), and sodium carbonate (Na_2_CO_3_) [14]. Researchers have reported ammonia as the most suitable alkaline solution [14]. Other researchers suggested ammonium hydroxide for alkaline pre-treatment and justified the suggestion as it is non-toxic, non-corrosive, and easy to retrieve [57]. It should be noted that, in alkaline pre-treatment, the operating conditions include low temperatures and pressures. In addition to these advantages, there is the possibility of the recovery of the reagent in excess alkaline solution [14]. As for disadvantages, there are fundamentally two, the high concentration of the alkaline reagent and the long reaction time [14]. Regarding alkaline pre-treatment, researchers [64] aimed to evaluate the effect of different types of pre-treatment on jute fibers (Corchorus sp.). For this, they tested four routes, i.e., mild alkaline pre-treatment (NaOH at atmospheric pressure); alkaline pre-treatment with bleaching (H_2_O_2_), alkaline pre-treatment with tetraacetylenediamine (TAED) and bleaching, and strong alkaline pre-treatment (NaOH at a pressure of 1 MPa). After the chemical pre-treatment, the defibrillation process was carried out by the grinding mechanical method. The best pre-treatment, alkaline with TAED and bleaching, reached a crystallinity index higher than 80% and produced CNFs with a diameter of 20 nm (Table 4).

As acid pre-treatments, hydrochloric, nitric, phosphoric, oxalic, and maleic acids, as well as some heteropolyacids (HPAs), are usually used [14]. In acid pre-treatment, the acid concentration, temperature, and cellulose:acid ratio are the parameters that determine the results [14]. A recent study consisted of evaluating two routes, acid (HCl) and enzymatic, for CNF preparation from wheat straw [65]. In that study, it was noted that the first stage was alkaline pre-treatment (in two stages with NaOH) for the removal of lignin and hemicellulose. Then, the CNFs were prepared by acidic and enzymatic routes. After the preparation of the CNFs by both routes, the mechanical homogenization treatment was applied. Regarding treatment with HCl, the authors reported a mild treatment when compared with the treatments with sulfuric acid or TEMPO, but that promoted subsequent defibrillation. It was concluded that the CNFs produced by the two routes, exhibited good thermal stability and identical yields. However, the CNFs produced by acid showed higher homogeneity and higher crystallinity. The CNFs produced with acidic and enzymatic pre-treatment had lengths and crystallinity indexes of 514 nm and 70%, and 1 μm and 48%, respectively (Table 4).

The organosolvation method is considered an emerging method that uses organic solvents, such as ethanol, ethylene glycol, and acetone, among others, to depolymerize lignin and hemicellulose, and thus make it easier to access cellulose [57]. In a study, four pre-treatments were used to prepare CNFs from the fruit of the silk wire (*Ceiba speciosa*). These pre-treatments were as follows: sodium hydroxide, sodium chlorite, a mixture of sodium hydroxide and hydrogen peroxide, and also a mixture of methanol with toluene [66]. After these pre-treatments, mechanical treatment and refining (10 passages) followed. The CNFs thus prepared were used to produce polymeric composites of polyvinyl acetate (PVAc). The fibers treated with sodium chlorite and those treated with a mixture of methanol and toluene were considered to be the best for the reinforcement of PVAc composites.

Oxidation aims to transform hydroxyl reactive groups of cellulose into carboxyl groups, aldehydes, and ketones [15]. As oxidizing agents, TEMPO is used, as well as ammonium persulfate (APS) [15] and sodium persulfate (SPS) [61].

TEMPO-mediated oxidation is a route in which this radical oxidizes the hydroxyl groups of cellulose and transforms them into carboxyl groups. In this context, a study on TEMPO-mediated oxidation [18] stated that it is possible to prepare nanocellulose from wood pulp with different morphologies, starting with oxidation by TEMPO, simply by controlling the experimental conditions (mechanical disintegration in water). the article stated that it was possible to prepare partially fibrillated nanonetworks, fully individualized cellulose nanofibers (CNFs) with a high aspect ratio (length/diameter ratio), and cellulose nanocrystals (CNCs) with a low aspect ratio using the most common TEMPO, which is TEMPO/NaBr/NaClO, in water at pH 10. In this study, NaClO was used as an oxidant consumed during oxidation, while TEMPO and NaBr acted as catalysts [18]. After being prepared, the TEMPO-CNFs were dispersed in water and subjected to ultrasound; thus, the CNCs were obtained. In that publication, it was mentioned that, in future studies, cellulose nanocrystals obtained by TEMPO will be developed—TEMPO-CNCs. Table 4 shows the values of the diameter, length, and aspect ratio of the various nanocelluloses obtained in that study.

While TEMPO-CNCs will be developed in the future, cellulose nanofibers obtained by oxidation mediated by TEMPO, the so-called TEMPO-CNFs, are already well-developed. In one publication, TEMPO-CNFs were prepared, which were then modified by amination with octadecilamine (ODA) [67]. In that work, bleached eucalyptus kraft pulp (95 to 98% cellulose) was used for nanofiber production, and the TEMPO/NaBr/NaClO system was applied in water at pH 10. After this pre-treatment, the mechanical treatment was performed, which consisted of the application of ultrasounds (20 min at 50 W). Then, the modification was conducted with ODA, followed by the formation of the film. The prepared films exhibited oil-repellence and anti-fingerprint properties, so they could be applied, for example, as smartphone protectors, because they can prevent the lack of visibility caused by fat and fingerprints. The CNF thus prepared had a diameter of about 3 nm and a length of 5 to 15 nm (Table 4).

Despite the existence of several studies with oxidation mediated by TEMPO, there are researchers who claim that this method has several disadvantages, i.e., the use of toxic reagents, limited oxidation, and the excessively long time of the steps [61]. These researchers suggest another agent as a strong oxidant, ammonium persulfate (APS), and report that it may be an alternative to TEMPO for CNC preparation for various reasons, such as its high solubility, low cost, and reduced toxicity. These researchers prepared CNCs of lemon (*Citrus limon*) seeds by three different routes, i.e., hydrolysis with sulfuric acid, oxidation with APS, and also oxidation with TEMPO. After this pre-treatment stage, mechanical treatment was followed by the application of ultrasounds (300 W for 30 min). In that study, the researchers verified that the CNCs produced via TEMPO exhibited a larger size (length from 340 to 380 nm) and lower crystallinity index (66.14%) than those produced by PHC (length from 140 to 160 nm and crystallinity index of 74.40%) (Table 4). According to the authors of that study, CNCs obtained by hydrolysis with sulfuric acid and those obtained by oxidation with APS could be used as emulsion stabilizers.

In the search for the best oxidation route, whether it be TEMPO or APS, some studies have been conducted. In one of these studies, bleached long-fiber kraft cellulose was used, made of Fir wood, for CNF production [10]. After pre-treatment by TEMPO and APS (ammonium persulfate), the mechanical treatment was applied via high-pressure homogenization (three passages at 300 bar, followed by three passages at 600 bar and three passages at 900 bar). The study concluded that oxidation by APS is cheaper than time-based oxidation for CNF production with the same degree of oxidation. However, it warned that cellulose nanofibers (CNFs) obtained by APS moderation cannot be hardened, as they otherwise become nanocrystals (CNCs). Another conclusion of this study was that the water retention value of CNFs produced with APS (APS_CNF) was higher than that of those produced with TEMPO (TEMPO_CNF), which may be useful for some applications. In addition, nanofibers prepared with PHC had a lower degree of polymerization (Table 5).

Another route for oxidative pre-treatment is sodium persulfate (SPS). In this context, a group of researchers produced CNFs from bleached softwood kraft pulp, combining SPS with ultraviolet (UV) light [17]. After this pre-treatment, high-pressure homogenization was carried out. The SPS method with UV proved to be effective in reducing the degree of polymerization and in transforming the hydroxyl groups into carboxyl groups. The CNFs obtained with SPS and UV exhibited a smaller diameter (<60 nm) and the highest crystallinity index of 81.31% (Table 4). The authors of that study stated that SPS with UV is a “green” route that produces CNFs with high mechanical resistance, so they can be useful as reinforcement in biopolymers.

Another pre-treatment is ozonolytic; this pre-treatment can be performed with ozone, hydrogen peroxide, or oxygen. It is an effective method for the removal of lignin [14]. According to the authors of that publication, ozone is the best agent to remove lignin because it attacks the carbon–carbon double bonds of aromatic compounds in a moist medium, and can this remove about 50–60% of lignin. An ozonolytic pre-treatment was used to prepare bamboo CNFs (*Bambusa chungii*) using nitric acid and hydrogen peroxide (via cooperative mechanism) under moderate conditions as a pre-treatment [5]. After the pre-treatment, high-pressure homogenization was performed (1500 bar), which took place without clogging and was achieved with only five passages. The CNFs thus prepared had a diameter of 13.1 ± 2.0 nm and high crystallinity index (74%), and exhibited excellent UV resistance and high thermal stability, so they could be applied in coatings (Table 4). The researchers who carried out that study reported that it was a simple, economic, and green route, and stated that the residual liquid from CNF preparation could be used as fertilizer.

Ionic liquids (ILs) are an emerging class of solvents. These are molten salts, i.e., they are liquids at room temperature. Ionic liquids do not ignite, are thermally stable, are recyclable, and have low steam pressure [1]. This set of properties makes them very attractive for various applications, and also for the extraction of nanocellulose. Ionic liquids are designed to dissolve certain solvents, and an ionic liquid’s design is based on three pillars of complex interactions—the Coulomb Law, van der Waals forces, and hydrogen bonds [21]. In a review on this subject, the following sequence of ionic liquids was recorded: the basis of imidazolium > pyridium > ammonium > carboxylate > alkylphosphate > halide-based Ils. This sequence is the descending order of their ability to dissolve cellulose [1]. In the context of the extraction of cellulose and ionic liquids, another recent review stated that ionic liquids are suitable for performing mechanical electrospinning treatment and indicated several applications for CFN obtained via this route [21]. In that review, it was mentioned that about one-sixth of the published papers on the extraction of cellulose fibers used ionic liquids. One study used ionic liquids for the production of CNFs from cotton linter [68]. In that study, hydrolysis was performed using an ionic liquid, the h of 1-butyl- 3-methylimidazolium hydrogen sulfate ([Bmim] H_2_SO_4_), and then a mechanical treatment was applied by homogenization at high pressure (25–100 Mpa, 10–60 min). The CNF thus produced had a diameter of 50 to 100 nm and a length of 500 to 800 nm (Table 4).

Recently a group of researchers proposed an innovative route for the production of CNFs from wood [16]. This was a very original work, because the researchers proposed, instead of the traditional route with seven stages, a new route with only three stages. In the traditional route, the seven stages are pulping (1), washing (2), bleaching (3), washing (4), TEMPO oxidation (5), washing (6), and mild shearing (7). In the new route, the three steps are simultaneous bleaching and TEMPO oxidation (1), washing (2), and mild shearing (3). This new route is promising because, as the authors pointed out, it has the following characteristics: simple, fast, cheap, requires less energy, generates less residual water, and still proves to be efficient for the production of CNFs. It should be emphasized that the diameter obtained for cellulose nanofibers in the traditional route was 6 ± 3 nm, and that via the new route was 5 ± 3 nm (Table 4). Compared with the traditional route, the thermal, optical, and mechanical properties that the CNFs obtained via the new route exhibited were similar.

**Table 4 materials-16-03104-t004:** Pre-treatments and/or chemical treatments (2019–2021).

Source	Treatments and Pre-Treatments	Some Properties of Nanocelluloses	References
Rami *(Boehmeria nivea)*	H_2_SO_4_ acid hydrolysis (after chemical purification)	CNC Diameter = 6.67 nm Length = 145.61 nm Crystallinity Index-90.77%	[58]
Cotton	Acid hydrolysis H_2_SO_4_	CNC Length-28 to 470 nm Crystallinity Index-55.76 ± 7.82%	[62]
	Acid hydrolysis H_2_SO_4_ with ultrasounds (60 min, 120 W)	CNC Length-10 to 50 nm Crystallinity Index-81.23%	[62]
Jute (*Corchorus sp.)*	Alkaline pre-treatment + TAED+ bleaching + grinding	CNF Diameter = 20 nm Crystallinity Index->80%	[64]
Wheat straw	Alkaline pre-treatment (two-stage) + Acid pre-treatment (HCl) + Homogenization	CNF Diameter ≈ 17 nm Length = 514 nm Crystallinity Index-70%	[65]
	Alkaline pre-treatment (in two stages) + Enzymatic pre-treatment + Homogenization	CNF Diameter ≈ 17 nm Length = 1 μm Crystallinity Index-48%	
*Ceiba* *Speciosa fruit fiber*	Organosolvation with Sodium Chlorite + Refining (1) Organosolvation with mixture of toluene and methanol + refining (2)	The CNF produced by the pathways (1) and (2) were considered as reinforcement of the PVAc composites.	[66]
Wood	TEMPO/NaBr/NaClO in water at pH = 10 + ultrasound (60–120 min)	Different morphologies: -nanonetworks -nanofibers: individualized Diameter ≈ 3 nm; Length > 500 nm Aspect ratio > 150 -nanocrystals: Diameter ≈ 3 nm; Length of 150–200 nm; Aspect ratio ≈ 50	[18]
Bleached eucalyptus kraft pulp	TEMPO/NaBr/NaClO in water at pH = 10 octadecilamine (ODA)+ ultrasounds (20 min, 50 W)	CNF Diameter ≈ 3 nm Length-5 to 15 nm	[67]
Lemon (*Citrus limon*) seeds	Hydrolysis with H_2_SO_4_ + ultrasound (30 min, 300 W)	CNC Diameter = 12–25 nm; Length = 130–170 nm; Crystallinity Index-69.67%	[61]
	Oxidation, APS + ultrasounds (30 min, 300 W)	CNC Diameter = 10–20 nm; Length = 140–160 nm; Crystallinity Index-74.40%	
	Oxidation, TEMPO + ultrasounds (30 min, 300 W)	CNC Diameter = 26–42 nm; Length = 380 nm; Crystallinity Index-66.14% = 340-	
Bleached kraft softwood pulp made from spruce	APS oxidation + High pressure homogenization (300 bar, 600 bar, 900 bar)	CNF Polymerization degree = 475 ± 15 Water retention value = 1.7 ± 0.11 (g H_2_O/g) Diameter = 20–40 nm	[10]
Bleached softwood kraft	Oxidation by SPS High pressure homogenization (100 MPa, at 5 °C)	CNF Diameter < 80 nm Crystallinity index-70.57%	[17]
	SPS oxidation + UV High pressure homogenization (100 MPa, 5 °C)	CNF Diameter < 60 nm Crystallinity Index-81.31%	
Bamboo (*Bambusa chungii*) Three-year-old	Nitric acid + Hydrogen peroxide + High pressure homogenization (1500 bar)	CNF Diameter = 13.1 ± 2.0 nm Crystallinity Index-74%	[5]
Cotton linter	Ionic Liquid [Bmim] H_2_SO_4_ High pressure homogenization (25–100 MPa, 10–60 min)	CNF Diameter = 50 to 100 nm Length = 500 to 800 nm	[68]
Wood	Bleaching and TEMPO (simultaneously) No pre-treatment	CNF Diameter = 5 ± 3.0 nm Diameter = 6 ± 3.0 nm	[16]
Sugarcane bagasse	Chemical pre-treatment (with sodium chlorite and KOH) + Enzymatic pre-treatment (mixture of three enzymes) + Ultrasounds (250 W, 20 min)	CNF Diameter = 7± 3.0 nm Crystallinity Index-65% and 72% (for 24 h and 3 h, incubation, respectively)	[69]
Poplar wood	Steam explosion + sodium chlorite + ultrasound-assisted enzymatic hydrolysis (200 W, 20 min)	CNF Diameter = 20 to 50 nm Crystallinity Index-61.98%	[70]

**Table 5 materials-16-03104-t005:** Some characteristics of APS_CNF and TEMPO_CNF (adapted from [11]).

CNF Type	Degree of Polymerization	Water Retention Value (g H_2_O/g)
APS_CNF	475 ± 15	1.7 ± 0.11
TEMPO_CNF	770 ± 21	1.3 ± 0.06

### 3.3. Enzymatic Method

Enzymatic hydrolysis is an environmentally friendly alternative to CNF preparation [13,15]. Specific enzymes, xylanases and ligninases, degrade hemicellulose and lignin, but maintain cellulose. Cellulases, on the other hand, hydrolyze cellulose [13]. There are three types of cellulases, namely endoglucanases (EGs), cellobiohydrolases (CBHs) or exoglucanases, and β-glucosidases (GBs) [15]. Each type of cellulase has a specific action: EG acts in the β-1,4-glycosidic links, CBH acts on the final part of the linear cellulose chain in order to degrade the crystalline zone of the cellulose, and GB hydrolyzes the cellulose into glucose [15]. It should be noted that, when the goal is to use cellulase enzymes to prepare cellulose nanocrystals, they must be removed from the mixture of enzymes, the CBHs [16]. The main disadvantages of enzymatic hydrolysis are the long operating time and low yield [15].

In one study, CNFs were prepared from poplar wood with the following steps: steam explosion, delignification (with sodium chlorite), and ultrasound-assisted enzymatic hydrolysis (200 W for 20 min) [70]. In that study, some operational parameters were evaluated, namely the cellulase dose, enzymatic hydrolysis time, and operating temperature. The researchers found that, for all of these parameters, there was an optimal value from which an increase in that parameter caused a decrease in yield. The optimum values found were a cellulase dose of 200 U/g, time of 12 h, and temperature of 50 °C, which corresponded to a yield of only 13.2% (Table 4). Recently, CNFs were prepared from sugarcane bagasse [69]. In that study, a mixture of enzymes was used to perform enzymatic hydrolysis; the mixture consisted of endoglucanases, xylanases, and lytic polysaccharide monooxygenases (LPMOs). First, a chemical pre-treatment was performed with sodium chlorite and potassium hydroxide; then, enzymatic hydrolysis was performed with the mixture of enzymes, and, finally, the mechanical treatment was performed that consisted of the application of ultrasound (250 W for 20 min). In that study, another route was also tested in which enzymatic treatment was replaced by TEMPO. From the comparison between the two routes, the researchers concluded that the enzymatic route took longer to prepare the CNFs, but the CNFs thus produced were more thermostable (resistance up to 260 °C) (Table 4).

## 4. Brief Reference to Functionalization

Cellulose has three hydroxyl groups in each glucose unit, which give it chemical reactivity that is favorable to the introduction of functional groups [2] This introduction of functional groups to cellulose is called functionalization. It is a strategy to modify and improve the properties of nanocellulose. For example, nanocelluloses can be functionalized to display the selective absorption property of contaminant ions, which can be applied in membranes for the treatment of water [11].

A review article has recently been published on the framework of functionalization [14]. In that review, three functionalization routes were described in detail, including functionalization to confer ionic surfaces (phosphorylation, carboxymethylation, sulfonation, and oxidation), functionalization to generate hydrophobic surfaces (acetylation, etherification, silylation, urethanization, and amidation), and functionalization by polymer grafting. In that recent review, three routes are described (“graft of”, “graft to”, and “graft through routes”). Regarding the graft through routes, in the review, four processes were described as well as their respective mechanisms, namely the atomic transfer radical polymerization route of grafting (ATRP), reversible addition-fragmentation chain transfer route of grafting (RAFT), free radical grafting route of functionalization, and the ring-opening polymerization route of grafting (ROP) [14]. In this review, only this brief reference is made to functionalization; to a reader interested in furthering this theme, it is advisable to consult the aforementioned review. However, in Section 7.4, a strategy for the functionalization of nanocellulose is explored, albeit slightly, in the case of triboelectric nanogenerators (TENGs).

## 5. Some Properties of Nanocelluloses

As nanocelluloses originate from cellulose, they have a very interesting property in that they are renewable, and can even be claimed to be inexhaustible, as the annual pulp production is 1.5 × 10^12^ tons [2]. In addition to this property, nanocelluloses exhibit a set of unique properties, such as biodegradability, low density, high strength and stiffness, high surface area, and low thermal expansion [9]. The density of nanocelluloses is low, so they are light materials (1.6 g/cm^3^) [12,21,23]. Within nanocelluloses, CNCs that, due to their high crystallinity index, manifest a set of very attractive properties stand out, such as mechanical properties (high tensile strength and high stiffness), thermal properties (high thermal stability and reduced coefficient of expansion), rheological properties (high storage modulus, for example), optical properties (high transparency), and peculiar physical–chemical properties, such as high permeability to gases [22]. This review aimed to gather the numerical values of some of the properties of nanocelluloses (Table 6). However, it should be noted once again that these values vary depending on the pulp source, the production method, and the operational parameters. Thus, regarding thermal degradation, there have been publications that reported that CNCs began to degrade above 310 °C, and CNFs at 270 °C [12]. Regarding the surface area of CNCs, there are publications that state values between 50 and 200 g/m^2^ [3] **,** and others that indicate a surface area of about 150 g/m^2^ [39].

Cellulose nanofibers can act as a barrier to oxygen, and in this context, the value of 17–18 mL/m^2^/day has been indicated for 0% relative humidity [3]. For BC, this property has fairly high values of 415.27 and 1962.67 mL/m^2^/day [24].

Other properties of nanocelluloses that make them very attractive are their elastic modulus and tensile strength. For CNCs, the literature reports elastic modulus values between 110 and 220 GPa, which, at a lower limit, are similar to that of Kevlar (124 to 130 GPa) and higher than that of steel (210 GPa) [3,12]. For CNFs, the elastic modulus is about 100 GPa [12]. Nanocelluloses thus have elastic modulus values much higher than those of native cellulose, the elastic modulus value of which is 5 to 30 GPa [19].

Regarding tensile strength, researchers have reported that CNCs have strengths of 7.5 to 7.7 GPa, which are much higher than those of Kevlar (3.5 GPa) and steel (4.1 GPa) [3,71]. Other researchers claim that CNCs exhibit an elastic modulus of about 10 GPa [39] which is about eight times that of steel [23]. Regarding biocompatibility, BC stands out. Materials made with BC are biocompatible and, therefore, have numerous biomedical applications [20]. It should be noted that biocompatibility is afforded by the ability of a foreign material to be implanted in the human body without causing reactions [20]. Another property of BC is to function as a UV radiation filter, presenting the following transmittance values: from 2.48 to 5.61% in the UVA region, from 0.33 to 2.10% in the UVB zone, and from 3.75 × 10^−3^ to 4.95 × 10^−1^% in the UVC region [25].

The developments in the routes and functionalization allow the chemical modification of nanocelluloses to display the desired properties.

**Table 6 materials-16-03104-t006:** Numerical values of some nanocelluloses’ properties.

Properties	Description	Reference
Thermal degradation	CNFs initiate thermal degradation at 270 °C CNCs initiate thermal degradation above 310 °C	[12]
Surface area	CNCs have a high surface area, from 50 to 200 g/m^2^ CNCs have a surface area of about 150 g/m^2^	[3] [39]
Oxygen barrier	CNFs have an oxygen transmittance rate, at 0% humidity, of 17–18 mL/m^2^/day BC presents values of 415.27 and 1962.67 mL/m^2^/day	[3] [24]
Tensile strength	CNCs exhibit tensile strength values from 7.5 to 7.7 GPa CNCs have about 10 GPa of tensile strength	[3,71] [39]
Elastic modulus	CNCs display elastic modulus values from 110 to 220 GPa For CNFs, the elastic modulus is about 100 GPa	[3,12] [12]
UV barrier	BC presents the following transmittance values: 2.48 to 5.61, 0.33 to 2.10, and 3.75 × 10^−3^ to 4.95 × 10^−1^% in the UVA, UVB, and UVC regions, respectively	[24]
Density	Nanocellulose is ligth and has a density of 1.6 g/cm^3^	[12,21,23]

## 6. Various Applications of Nanocellulose

The properties of nanocelluloses (referred to in Section 5) make these nanomaterials attractive for numerous applications. In this context, numerous studies and patent registrations have been developed. A recent study on patents for nanocelluloses showed that, between 2010 and 2017, 4500 patents were registered and 70% of the 4500 patents were published between 2015 and 2017 [9]. The countries where the most patents were registered were China, the United States, and Japan [9]. The research also mentioned that, in relation to the number of for-profit companies, about 50% of the total patents involved the production of bacterial cellulose (BC) and cellulose nanocrystals (CNCs). For the production of cellulose nanofibers (CNFs), this number was much higher and accounted for around 80% of the total patents.

Next, innovative applications of nanocelluloses are presented (Table 7). The versatility of nanocelluloses that encompasses applications in different areas, such as cosmetics, electronics, medicine, pharmaceuticals, packaging, food, electrochemical paper-based analytical devices (ePAD), water treatment, and environmental remediation, should be emphasized.

Another application of nanocelluloses is in the production of triboelectric nanogenerators (TENGs), but given the revolutionary character and the profound implications that are predicted from the application of TENGs, this application will be described in Section 7 of this review.

Recently, an article describing the production of a biosensor made of bagasse pulp CNFs, and DNA graft (CNF-DNA) for detecting silver ions and acetylcholinesterase (AChE) was published [72]. This biosensor allowed the detection of Ag^+^ ions at very low concentrations, in the order of 10^−6^ nm, and in the presence of other metal ions, such as Hg^2+^, Ba^2+^, Cd^2+^, Mg^2+^, Pb^2+^, and Zn^2+^. It also allows measuring AChE at residual levels (0.053 mU/mL) and in the presence of reagents that interfere with AChE. This biosensor is very important in water control and medicine, as it is known that a concentration of silver ions above 1.6 nmol/L is harmful to fish, and that AChE is a central neurotransmitter, which is related to Alzheimer’s disease. It should be emphasized that, to obtain the CNF, oxidation by TEMPO and homogenization (four passages in the homogenizer) were performed.

Pills that disintegrate easily are a goal of the pharmaceutical industry. In a recent investigation, super-disintegrant tablets were prepared with CNCs prepared from microcrystalline cellulose through acid hydrolysis; then, a functionalization was performed with 2-hydroxy-ethyl methacrylate (HEMA)/itaconic acid (IA). The results revealed that disintegration was faster for CNC tablets (below 1 min) than for traditional tablets [73].

Silicon seems to be an appropriate material to serve as the anode in lithium-ion batteries; however, it has the inconvenience of changing volume, which affects battery performance. Recently, an auxiliary additive for silicon anodes was produced consisting of TEMPO-oxidized cellulose nanofibers (CNF-TEMPO) [74]. Researchers have stated that CNF-TEMPO hydroxyl groups establish hydrogen connections with silicon, and that it only takes about 1% (by weight) CNF-TEMPO to improve battery performance.

In another study, willow bark was used to produce CNFs for a film. Generally speaking, first, an extraction was performed with hot water; then, the treated bark was passed through a microfluidizer to obtain the CNFs, and treatment with *p*-toluenossulfonic acid was then performed [75]. These films could act as a barrier to oxygen and UV radiation, so they are promising in food packaging, once the oxidation induced by the passage of radiation is decreased.

Innovative adhesives with dissolvable hyaluronic acid (HA) microneedles, combined with rutin (vitamin P), have recently been designed [76]. They are used as anti-aging cosmetics. In this cosmetic, the base is the pore structure of BC, where HA and rutin are placed. BC thus works as a support for the incorporation of these active ingredients. The BC used in this investigation was from the *Gluconacetobacter sacchari* bacteria family. It was found that rutin introduced in BC maintained its antioxidant activity for 24 weeks. This cosmetic was applied to volunteers, and none presented irritation.

Another application of nanocelluloses is as a food additive for biscuit cream. An innovative work consisted of two parts. In the first part, cotton linter CNCs were prepared—hydrolysis with sulfuric acid was performed, followed by homogenization with ultrasound (300 W, 4 min) [77]. CNCs with 89 nm diameter were obtained. In the second part, the CNCs were incorporated into biscuit cream (5% by weight of CNC). After 24 h of the preparation of the biscuits, sensory evaluation was performed by twelve people, and the results were good.

Nanocelluloses have the ability to stabilize emulsions; for this reason, they are used in foams, puddings, salad dressings, etc. [57]. CNCs are widely used in feeding because they are able to stabilize emulsions and act as thickeners [78].

Recently, a CNF aerogel modified with three-dimensional polyethylene imine (PEI) (cylindrical 3-D) was produced (PEI@CNF) [79]. Cu^2+^ ions are known to be a wastewater pollutant, and the EPA (Environmental Protection Agency) states that their concentration in tap water must be less than 1.3 mg/L. In this context, a PEI@CNF has been produced that can continuously remove Cu^2+^ ions from wastewater. The cellulose nanofibers (CNFs) used in this aerogel were obtained from hardwood pulp and had a diameter of 50 nm.

Films composed of chitosan and CNCs were produced for the preparation of meat packaging [80]. In that study, the CNCs were acquired from a company and had 75 nm diameters. The researchers claimed that CNCs improved the thermal stability and resistance of films, and also found that they constituted a barrier to oxygen absorption. Several CNC concentrations were tested, and it was concluded that the incorporation of 10 to 25% (by weight) achieved higher antimicrobial activity in chicken meat packaging. This example used CNCs, while the previous example used CNFs.

The development of electronics involves all-solid-state supercapacitors (ASCs). One of the components of ASCs is the hydrogel electrolyte. A hydrogel electrolyte with a composite based on bacterial cellulose (BC) and polyacrylamide (PAM) was produced [81]. The researchers claimed that BC nanofibers offered mechanical resistance without compromising flexibility and simultaneously served as channels for transporting ions. The ASCs exhibited high ionic conductivity (125 mS cm^−1^), high tensile strength (330 GPa), and super elasticity (extensibility of up to ≈1300%).

Recently, researchers dried BC in the air and claimed to have thus created small-caliber artificial blood vessels (<6 mm) [82]. Although there were already artificial blood vessels made from BC-type hydrogel (BC-GEL), they had some drawbacks, one of which was the difficulty to saturate during surgery. To overcome this difficulty, the researchers dried the BC in air, creating dry BC vessels of 3 mm. These dry BC vessels were tested in the replacement of the carotid artery of rabbits, and the results were good. In that study, *Komagataeibacter xylinus* (formerly *Gluconacetobacter xylinus*) was used.

An electrochemical paper-based analytical device (ePAD) has recently been produced for glucose measurement in blood samples [83]. This ePAD was based on CNFs prepared by electrospinning cellulose acetate, and then chemically modified by deacetylation in an alkaline solution. This device revealed good sensitivity and high reproducibility.

Recently, a solid sensor for cannabis detection was developed. In this sensor, the solid substrate was TEMPO-oxidized CNF. In the production of this sensor, a microfluidizer was also used at 1800 bar pressure, through which the suspension was passed twice [84].

Another recent work consisted of reinforcing polyurethane with nanocellulose for cork floors. The study revealed that the introduction of nanocellulose increased the elastic deformation of cork, a property that is important for its use as a flooring material. In this work, two types of nanocellulose were used, poplar wood nanocellulose and bacterial nanocellulose, to which individually water-based polyurethane (WPU) was added. There were considerable increases in the modulus of elasticity of 54.50% and 120.48% compared with that of pure WPU (312.74 MPa), respectively [85].

As a wound dressing, an investigation produced a banana stem nanocellulose aerogel for wound dressing. In that study, bleaching was performed with sodium chlorite (pH = 4). Afterward, alkaline treatment with potassium hydroxide was carried out, along with ultrasonic grinding [86].

In another investigation, dressings for wounds were produced with a bacterial nanocellulose hydrogel, to which mesoporous silica nanoparticles loaded with a pH-sensitive dye were added, which made it possible to continuously read the pH of the wound without having to remove the dressing, facilitating treatment [87].

Recently, an optical phenanthroline–nanocellulose (5-Phen) sensor was developed, which allowed the rapid detection of Fe(II) and Pd(II) ions (less than 1 min) based on the spectrophotometric technique. For the production of nanocellulose, commercial cellulose was used, which was hydrolyzed with sulfuric acid; then, a suspension was obtained that was washed with deionized water, and the powders obtained by centrifugation were ground [88].

Another recent application of nanocelluloses was the production of a hybrid of cellulose nanocrystals (CNCs) with CuO (II), which proved to be a good lubricant additive for engine oils, as it could prevent piston wear [89].

Recently, a bacterial nanocellulose biofilter was developed to remove microbes from water. Nanocellulose was obtained from pineapple peel waste. In the investigation, the amount of microbes was measured in the water before and after filtration, and it was found that the amount of microbes decreased by about 50% [90].

In another investigation, nanocellulose derived from corn straw proved to be effective as an oil-in-water emulsion stabilizer. These CNFs, with a width of 4 nm and length of 353 nm, were obtained via oxidation by TEMPO [91].

Recently, CNC multifunctional aerogels were produced, to which calcium chloride was added as a green crosslinker, and then lyophilized. These aerogels showed great broadband acoustic absorption capacity, as they exhibited high absorption coefficient values, the highest value being 0.99 (for a frequency of 2960 Hz) and the lowest being 0.8 (for a frequency of 4673 Hz) [92].

In a recent investigation, an electronic component made of copper and nanocellulose was produced. This component is electrothermal, flexible, and elastic; owing to these characteristics, it is expected to be applicable in wearable devices [93].

Several studies used nanocellulose to produce value-added chemicals, such as glucose and 5-hydroxymethylfurfural, via the catalytic processing of nanocellulose, such as hydrothermal hydrolysis of microcrystalline cellulose [94] or nano-fibrillated cellulose for the catalytic-transfer hydrogenation of 5-hydroxymethyfural to 2,5-bishydroxymethylfuran [95].

There have been numerous reviews on the applications of nanocelluloses, such as how to function as ingredients in food [96], biomedical uses and biosensors [97], in the development of hydrogels for smart medicines [98], for food packaging [24] **,** in organic light-emitting diodes (OLEDs) [99], and in membranes [100].

**Table 7 materials-16-03104-t007:** Some applications of nanocelluloses (published from 2021 to 2023).

Innovation	Constitution and/or Methods	Applications	Reference
Biosensor to detect silver ions and AChE	CNFs of bagasse pulp and DNA graft To obtain the CNFs, oxidation by TEMPO and homogenization were performed	Control of silver ions in water AchE control (Alzheimer’s disease-related parameter)	[72]
Quick-dissolving tablets	CNCs obtained by acid hydrolysis and then functionalized	Pharmaceutical industry	[73]
Assistance for Si anode in lithium-ion batteries	CNFs oxidized by TEMPO	Energy (lithium-ion batteries)	[74]
UV-blocking films	CNFs obtained from willow bark by microfluidization and acid treatment	Food packaging	[75]
Adhesives with microneedles	BC supported hyaluronic acid (HA) and rutin (vitamin P).	Anti-aging cosmetics	[76]
Cream for biscuits	CNCs obtained from cotton linter by acid hydrolysis (with H_2_SO_4_) followed by homogenization coupled with ultrasounds	Food	[77]
Aerogel PEI@CNF continuously removes Cu^2+^ ions	Aerogel made of hardwood pulp CNFs and PEI	Wastewater treatment.	[79]
CNC and chitosan films prevent degradation	CNC films (purchased from a company) and chitosan	Food packaging	[80]
All-solid-state supercapacitor (ASC)	BC composite and polyacrylamide (PAM)	Electronics—flexible energy storage	[81]
Artificial blood vessels	Air-dried BC (BC_DRY)	Medicine	[82]
Electrochemical paper-based analytical device (ePAD)	CNF obtained by electrospinning with cellulose acetate and then chemically modified	Medicine	[83]
Solid sensors and paper-based immunoassays by conjugating antibodies to nanocellulose	The solid substrate was TEMPO-oxidized CNF Microfluidizer at 1800 bar pressure (two passes)	Cannabis detection	[84]
Nanocellulose-reinforced polyurethane	Poplar wood nanocellulose and bacterial nanocellulose Oxidation by TEMPO	Flexible coating for cork floors	[85]
Nanocellulose aerogels (from banana stem)	Sodium chlorite bleaching, potassium hydroxide alkaline treatment, and ultrasonic crushing treatment	Wound dressing	[86]
Nanocellulose composite for real-time wound pH monitoring	Bacterial nanocellulose hydrogel with mesoporous silica nanoparticles and a pH-sensitive dye	Wound dressing	[87]
Phenanthroline–nanocellulose optical sensor	Nanocellulose obtained from commercial cellulose by hydrolysis with sulfuric acid	Rapid detection of Fe(II) and Pd(II) ions (less than 1 min)	[88]
Lubricant additive for oils	CNC and CuO (II)	Preventing engine piston wear	[89]
Bacterial nanocellulose biofilter from pineapple peel waste	Bacterial nanocellulose produced with the bacteria *Acetobacter xylinum*	Removal of microbes from water	[90]
Corn stover-derived nanocellulose as a stabilizer of oil-in-water emulsions	CNFs derived from corn straw, obtained by TEMPO oxidation and ultrasound	Emulsion stabilizer	[91]
Nanocellulose aerogels	For the preparation of aerogels, CNCs supplied by a company were used	Acoustic absorption (applications in construction, transportation, and environmental acoustics)	[92]
Electronic component of copper–nanocellulose	Did not mention how the nanocellulose was prepared	Conformal electronics	[93]

## 7. Triboeletric Nanogenerators (TENGs) Based on Nanocelluloses

Triboelectric nanogenerators (TENGs) are devices that convert mechanical energy, which is abundant, into electrical energy [2,101]. These devices, when applied, for example, in shoe insoles, allow the collection of mechanical energy from human movement and turn it into electrical energy [102]. In a study in which this was carried out, a maximum voltage and an output current of 210 V and 45 μA were achieved on a walk. This power would allow 214 light-emitting diodes (LEDs) connected in series to be ignited [102]. From this study, it is inferable that a TENG applied to clothing can collect the energy of human movements and turn it into electricity, and this electric energy will be enough to power wearable electronic devices, such as a watch or a fitness tracker [101].

TENGs have recently been the subject of several revision papers [2,101,103,104,105]. Such publications refer to the use of not only the energy of human movement, but also mechanical energy—mechanical vibrations, waves of the sea and wind, and even the movement of animals or the water of a tap [106]. Recently, a group of researchers built three TENGs based on rolling spheres that were introduced into a buoy to capture the energy of the waves of the sea [107].

TENGs are emerging devices for the beginning of the next era, the age of TENGs; in this era, there will be innovative applications, such as self-powered sensors, and wearable and implantable electronic devices [101]. Researchers assure that, with the development of TENGs, a revolution will occur, because it will move from passive sensors and those powered by a battery to self-powered sensors, and point out that this revolution is only comparable to the revolution where we moved from wired communication to wireless communication [101].

### 7.1. Operation and Constitution of Triboeletric Nanogenerators (TENGs)

A TENG is a device that works based on the electrostatic effect, contact, and electrostatic induction [105,108]. When a force (bending, elongation, or compression) is applied to two materials of different frictional polarities (tendency to gain or lose electrodes), the materials contact, and due to this contact, loads are generated on their surfaces. Once loaded, the materials are different, resulting in the creation of a difference in the induced electrical potential and, consequently, the production of an electric current (Figure 4). Studies indicate that the electrical output power of a TENG is directly proportional to the square of the load density; therefore, improving the performance of a TENG requires increasing the load density on its friction surface [108].

Historically, the first TENG was produced in 2012 and consisted of stacking two thin films of two different polymers, one Kapton, and the other PET, with the dimensions of 4.5 cm × 1.2 cm. The Kapton film had a thickness of 125 μm and the PET film had a thickness of 220 μm. Both surfaces of these materials were covered by a thin gold alloy coating (100 nm thick) by sputter coating. This device, when folded, generated a voltage of 3.3 V and an electric current of 0.6 μA [109].

**Figure 4 materials-16-03104-f004:**
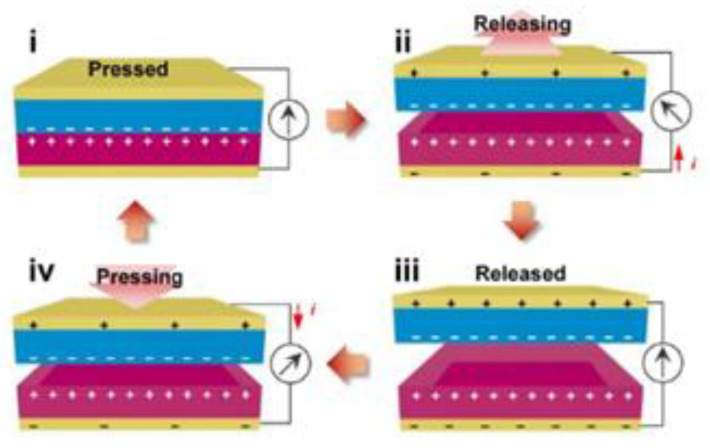
Scheme representing the general functioning of a TENG. [110], (**i**) Pressed (**ii**) Releasing (**iii**) Released (**iv**) Pressing.

Traditionally, TENGs are formed by layers of metal and polymer friction. The most commonly used metals are aluminum and copper [2]. As for polymers, the following are used: polyamide (PA), polydimethylsiloxane (PDMS) [2], FEP (fluorinated ethylene propylene), PTFE (polytetrafluoroethylene), PVDF (polyvinylidene fluoride), and PET (polyethylene terephthalate) [111]. It should be noted that a recent review assessed the potential of various polymers for their applicability in TENGs, with emphasis on self-healing polymers and those with memory [103].

In choosing the polymer for the construction of a TENG, its position in the triboelectric series should be taken into account. It should be noted that, recently, researchers have ordered several polymers in a quantitative triboelectric series; this quantitative series is an important tool in the design of a TENG, because the further away the two materials to be used in the series, the greater their TENG performance [108].

### 7.2. Advantages and Disadvantages of Using Nanocellulose in TENGs

TENGs consisting of metals and synthetic polymers have several disadvantages. It is known that metals easily undergo corrosion processes, which affects the performance of TENGs [2]. As for polymers, the fact that they are synthetic creates dependence on oil, so it is urgent that renewable-based polymers are found. Thus, one of the current challenges in the area of TENGs is building devices with good performance, but on a renewable basis. It is in this context that nanocelluloses emerge as promising materials to be used as substrates in the construction of TENGs. A recent review on the application of nanocellulose in TENGs pointed out several advantages, including high surface area, high crystallinity, high mechanical strength, low density, and low coefficients of thermal expansion [2]. However, despite these advantages, cellulose has an almost neutral polarity, and occupies a positive, but only slightly, load position in the triboelectric series [111]. Thus, nanocelluloses have weak polarization and generate low surface loads, which, at the outset has, as a consequence, low performance compared with synthetic polymers [111]. The current challenge of nanocellulose-based TENGs is to find strategies to increase the triboelectric load density [2].

### 7.3. Strategies for Increasing the Performance of TENGs with Nanocelluloses

As nanocelluloses have weak polarization [111], as already mentioned, by electrification via contact, they can generate low surface loads; additionally, the electrical output power of a TENG is directly proportional to the square of its load density [108]. Thus, TENGs with nanocelluloses have reduced electrical output power. It is, therefore, important to increase the polarization of nanocelluloses in order to overcome this limitation. Strategies to overcome this limitation include surface chemical functionalization and physical doping [2].

#### 7.3.1. Chemical Surface

Methylation and nitration are two examples of functionalization that consist of introducing methyl and nitro groups into nanocellulose groups, respectively. This was the strategy adopted for the production of the first TENG in 2017 based on cellulose nanofibers (CNFs) [111]. In that study, a triboelectric nanogenerator was produced using methyl-CNF and nitro-CNF films. The values of ethylene (Voc) and short-circuit current (Isc) reached were, respectively, 8 V and 9 μA [111].

In that study, the methyl-CNF and nitro-CNF films exhibited load density values of 62.5 and 85.8 μC m^−2^, respectively, which were considerable increases compared with the value of the pure CNF (13.3 μC m^−2^) [111] (Table 8). In that study, the CNFs were obtained from the bleached pulp of kraft eucalyptus, which were oxidized by TEMPO and then submitted to high-pressure homogenization through a set of chambers and then a microfluidizer three times [112].

Another TENG was made with CNF using amino functionalization. In this work, one of the triboelectric layers was an FEP film (fluorinated ethylene propylene) and the other layer was CNFs modified by polyethyleneimine (PEI) reticulated with glutaraldehyde, which was then coated with Ag nanoparticles. The values of the open circuit voltage (Voc) and short-circuit current (Isc) reached were, respectively, 286 V and 4 μA [113] (Table 8). In that study, the CNFs were obtained by an ultra-thin grinder; then, high-pressure homogenization was performed in two chambers, the first with a large diameter at 350 bar, where 5 passages were performed, and the second at 1500 bar, where 10 passages were performed. Then, the amino functionalization was performed.

**Table 8 materials-16-03104-t008:** The first nanocellulose-based TENG [106]. Examples of TENGs based on CNCs, CNFs, and BC.

Pulp Polarization Strategy	Triboelectric Material 1	Triboelectric Material 2	Output Performance	Reference
Functionalization	Methyl-CNF film	Nitro-CNF film	Voc ≈8 V Isc ≈9 μA	[111]
Functionalization	CNF-PEI-Ag	FEP	Voc ≈286 V Isc ≈4 μA	[113]
Physical doping	PDMS/CNC composite film	Al	Voc ≈350 V 5 μA cm^−2^	[114]
Physical doping with poling	Paper composite and BC	PTFE film	Voc ≈170 V Isc ≈9.8 μA	[115]

#### 7.3.2. Physical Doping

Another strategy to increase the performance of nanocelluloses is physical doping. This method consists of introducing nanocelluloses with high dielectric constant into a material, such as particles of barium titanate or strontium titanate [2].

One of the studies that resorted to physical doping incorporated polydimethylsiloxane (PDMS) in the CNCs [114], and verified that the TENG performance increased 10 times relative to that of pure PDMS, and the open circuit voltage and short-circuit current density values reached, respectively, ≈350 V and ≈5 μA cm^−2^. This TENG exhibited, in continuous operation, an instant output power of 1.65 mW (Table 8). In this publication the authors did not fully clarify the process they used for the production of CNC, only reporting that they were produced by “wet ball milling and freeze-drying processes” [114].

TENGs produced using physical doping exhibit, after some time, certain wear and tear, because, as the TENGs operate on the basis of friction, this ends up wearing the materials introduced to increase the electrical load. Thus, TENGs produced using physical doping have low strength and, consequently, higher production costs and limited performance [108]. Chemical functionalization thus has advantages over physical doping.

In one paper, to increase the weak polarization of nanocelluloses, researchers used, in addition to physical doping, poling. Poling is a method that consists of applying, previously, an external electric field to promote the generation of loads on the surface [115]. In the study, the researchers used a composite of BC to build a TENG. This composite consisted of the BC matrix, silver nanowires, and nanoparticles of barium titanate (BaTiO_3_). This was one of the triboelectric materials; the other was PTFE. For this TENG, the researchers achieved an output voltage of 170 V and a current of 9.8 μA, which, according to the researchers, allowed 200 LEDs connected in series to be lit [115] (Table 8).

The four studies on TENGs mentioned above are summarized in Table 8. The existence of TENGs that use cellulose nanofibers (CNFs), other nanocrystals (CNCs), and bacterial cellulose (BC) should be highlighted. Therefore, it can be said that TENGs can be constructed from any of the nanocellulose morphologies.

The studies in Table 8 were selected from the literature because they reported high open circuit voltage (Voc) of 286 V for a TENG with CNFs [114], 350 V for one with CNCs, [114] and 170 V for a TENG made from BC [115] (Table 8).

### 7.4. Innovative Triboelectric Nanogenerators (TENGs) Based on Nanocelluloses

Table 9 presents the innovative TENGs based on nanocelluloses published in 2022 and 2023. Recently, a TENG was prepared from CNFs and retardants (black phosphorus and phytic acid). This TENG can warn of the possibility of a fire when changing temperature, with a response time of 5 s. In addition to this function, being a TENG, it can convert the kinetic energy of human movement into electrical energy (116 V and 3.02 mA·m^−2^ at 2 Hz). This TENG is promising as it can be wearable. The researchers of this work demonstrated that this TENG can also power an LED or charge an electronic clock [115].

In another recent investigation, the performance of a TENG was improved by the addition of CNFs (obtained from sugarcane bagasse), activated carbon nanoparticles, and natural rubber. With this TENG, a voltage of 137 V and a power density of 2.74 W/m ^2^ were achieved [116].

One of the problems associated with TENGs is the humidity that causes them to wear out. To combat this problem, researchers prepared TENGs with CNFs (from bagasse pulp) that were methylated and to which silica nanoparticles were added. This TENG allowed a voltage of 125 V to be obtained. This TENG can be used to transform the energy of ocean waves into electrical energy [117].

As mentioned in previous sections, CNCs can also be used in the preparation of TENGs. Recently, a TENG was prepared with a hydrogel formed by CNC, NaCl, and acrylic acid polymeric blocks. The result was an ultra-elastic TENG, which made it possible to detect human flexion movements (such as movements of the fingers, elbows, or wrists) and transform the energy of these movements into electrical energy. This TENG allowed a voltage of 89.2 V and power density of 60.8 mW m^−2^ at 1.5 Hz to be obtained. Owing to its good flexibility, this TENG can be used in wearable components [118].

Another promising work is the preparation of a TENG as a sweat sensor made from CNF hydrogels. This sensor can measure the presence of Na^+^, K^+^, and Ca^2+^ ions in sweat, and wirelessly transmit the results in real time to a mobile application. This TENG is self-powered by the energy of its user’s movement and allows monitoring physical exercise in real time, so it can be used in wearable components [119].

The presence of antibiotics in wastewater is an environmental problem. Recently, a TENG was prepared that could convert wave energy from an aeration tank into electrical energy and use this external electric field to promote tetracycline degradation. The researchers who tested this TENG in wastewater tanks stated that, when the water acceleration was 5 m s^−2^, in just 80 min, 95.89% of tetracycline was removed [120].

Researchers prepared a TENG to detect, in real time, human breathing. This TENG consisted of CNCs (extracted from waste printing papers) mixed with methylcellulose and graphite powder. After using the TENG, all of its components could be dissolved in water. This TENG allowed different degrees of breathing to be distinguished (light, normal, and accelerated) [121].

**Table 9 materials-16-03104-t009:** Innovative TENGs based on nanocelluloses (published from 2022 to 2023).

Some Constituents of TENG	Innovation	References
With CNFs and fire retardants (phosphorus and phytic acid)	Responds to the possibility of fire in 5 s, can be wearable, and can trigger LED light or an electronic clock	[122]
With CNFs (from sugarcane bagasse), activated carbon nanoparticles, and natural rubber	Power density of 2.74 W/m^2^ and highest voltage output of 137 V CNFs are extracted from agricultural waste (sugarcane bagasse)	[116]
With CNFs (from bagasse pulp) and silica nanoparticles	Superhydrophobic Can be used to transform the energy of ocean waves into electrical energy	[117]
With hydrogel formed by CNCs, NaCl, and acrylic acid polymeric blocks	Ultra-elastic TENG that detects human bending movements (fingers, elbows, or wrists)	[118]
Has CNFs obtained via TEMPO-mediated oxidation and ammonium persulfate (APS)	Self-powered sweat sensor that wirelessly transmits results to a mobile app in real time	[119]
CNFs and perfluoroethylene propylene film (FEP), and the triboelectric electrode is a sheet of Cu	In addition to converting wave energy into electrical energy, it uses electrical energy to promote tetracycline degradation Self-powered and promising system in wastewater treatment	[120]
CNCs (extracted from waste printing papers) mixed with methylcellulose and graphite powder	Totally soluble in water Allows monitoring of human respiration	[121]

### 7.5. The Age of TENG and Nanocelluloses

It is anticipated that the new era is the age of TENGs and, according to some authors, will be a revolution [101]. This revolution will allow applications that, a few years ago would be unthinkable, such as the creation of self-powered sensor networks and the mechanical energy of sea waves or wind; these in situ sensor networks may be useful in preventing forest fires or monitoring pollution, among other, numerous applications [101].

It is also expected that TENGs based on nanocelluloses will be developed [123]. There will be, for example, wearable, implantable, and skin-like electronic components that will allow the in situ quantification of physiological signals from the human body [123]. In short, it can be said that the era of TENGs is approaching and that nanocelluloses will certainly be promising materials in their structure.

## 8. Conclusions

Nanocelluloses exhibit a set of mechanical, thermal, optical, rheological, and physical–chemical properties that make them unique and promising, with applications in numerous areas. As shown in this review, the production of nanocelluloses involves mechanical, chemical, and enzymatic processes that influence, in a decisive way, the properties of nanocelluloses. Various types of chemical functionalization are discussed as strategies for modifying the properties of nanocelluloses.

Regarding the areas of application of nanocelluloses, only a few were presented in this review, including food, cosmetics, electronics, medicine, pharmaceutical, and water treatment. It should be noted, however, that this list of applications for nanocelluloses, although already extensive, is neither complete nor finalized. Furthermore, regarding the applications of nanocelluloses, the functioning of TENGs was presented in detail and some made with nanocelluloses were described. The age of TENGs is approaching, and those produced with nanocelluloses will certainly be the subject of further investigation.

Nanocelluloses are abundant, renewable, versatile, and biocompatible, which makes them emerging raw materials of crucial importance in a more sustainable future. This review thoroughly described the methods for obtaining nanocellulose in different forms such as CNCs, CNFs, and BC, and their preparation methods with insights from several very recent articles, and is expected to inspire the identification of new routes, preferably greener, for the production of nanocelluloses.

## 9. Challenges and Opportunities

There are numerous challenges in the production of nanocelluloses and their derivatives. The further prospects are mainly the transition from laboratory scale to industrial scale. In order for this to occur, the following challenges must be resolved:(1)Optimize the methods of preparing nanocelluloses, such as the development of optimal extraction and nanoprocessing methods that should be conducted in the future [124];(2)Improve the life-cycle assessment of materials with nanocelluloses [124];(3)Develop more environmentally sustainable preparation methods [100,124]. One of the problems in isolating nanocellulose is the recalcitrant nature of the biomass, which is why chemical methods are used, in which strong acid or base reagents are used, such as sulfuric acid (at 65%) and sodium hydroxide, leading to environmental problems [100]. Investing in cleaner technologies, such as ultrasound and microwaves, could help [100];(4)Develop economic industrial processes to market nanocelluloses for various applications. In this sense, the companies Diacel FineChem Ltd. (Japan) and Celluforce (Canada) are producing nanocelluloses on a large scale [100];(5)Using, from a bioeconomy and circular economy perspective, agricultural and forestry residues for the production of nanocelluloses [100], such as pineapple peel [90], banana stems [86], or corn straw [91].

## Figures and Tables

**Figure 1 materials-16-03104-f001:**
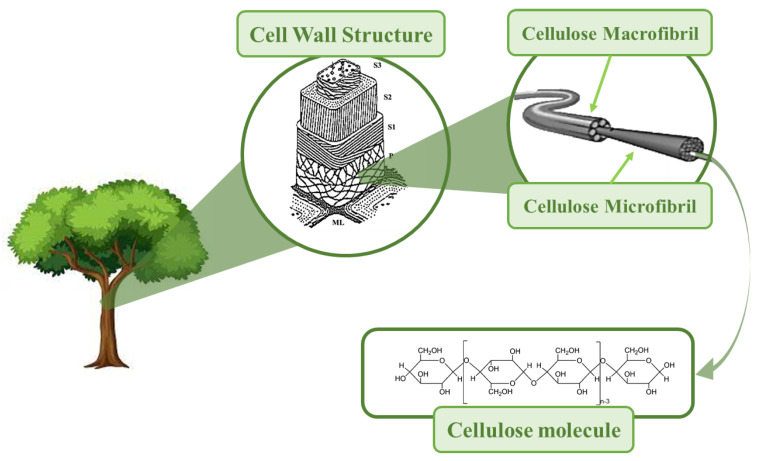
The simplified structural hierarchy of cellulose from a molecule to wood tissue.

**Figure 2 materials-16-03104-f002:**
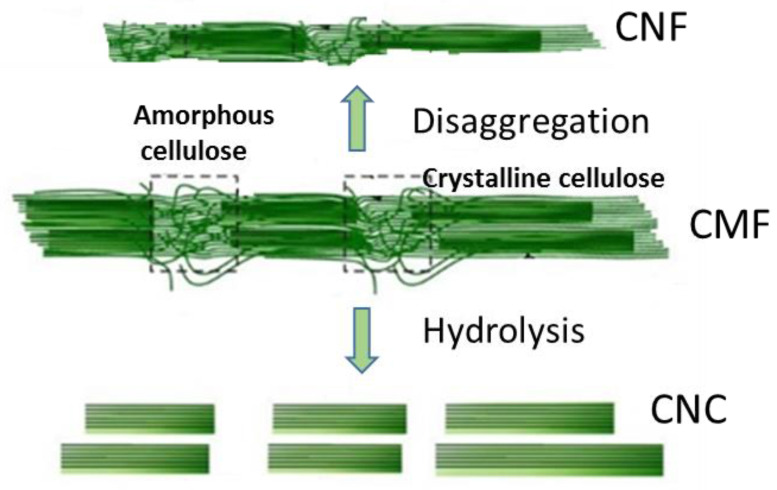
Schematic representation of cellulose nanofibrils (CNFs) production by disaggregation of cellulose microfibrils (CMFs) in the cell wall and the production of cellulose nanocrystals (CNCs) by the hydrolytic degradation of the amorphous part of CMF.

**Table 1 materials-16-03104-t001:** Medium size, sources, and preparation of the various cellulose nanomorphologies (adapted from [1,4,11]).

Approach	Nanomorphologies	Sources	Medium Size	Preparation
“Top-down”	Cellulose nanocrystals (CNCs)	Wood, cotton, hemp, linen, straw, tubers, tunicates, algae, bacteria, etc.	Diameter—5–70 nm Length—100 nm to several micrometers	Chemical treatment in the form of acid hydrolysis of cellulose (or enzyme-assisted hydrolysis)
“Top-down”	Cellulose nanofibers (CNFs)	Wood, cotton, hemp, beetroot, etc.	Diameter—5–60 nm Length—several micrometers	Defibrillation of wood pulp by mechanical treatment before and/or after chemical (and/or enzymatic) processes
“Bottom-up”	Bacterial nanocellulose (BC)	Low-molecular-weight sugars and alcohols	Diameter—20–100 nm Various kinds of nanofiber nets	Synthesized by bacteria

**Table 2 materials-16-03104-t002:** Studies on the operational parameters of mechanical methods for nanocellulose preparation.

Method and Description	Study of the Method in the Production of Nanocellulose	References
Refining Passage of cellulose pulp through two discs (one that rotates and one that is fixed)	Study of CNF properties during defibrillation in an ultra-disc refiner	[26]
High-pressure homogenization Passage of cellulose pulp into a pressurized valve	Study on the effects of refining and homogenization on the development of CNFs	[28]
Microfluidization Passage of cellulose pulp in a pressurized valve through Z-shaped channels	Study of microfluidization parameters (chamber size and number of passes) in CNFs	[29]
Grinding Passage of cellulose paste through two grinding stones (one fixed and one rotating)	Optimization of grinding (number of cycles) for CNF production after alkaline chemical pre-treatment	[31]
Ball milling Putting cellulose pulp in ball mill	Effects of different ball-grinding times on the microstructure and rheological properties of CNFs Study of multiple parameters (isolated or combined) of ball-grinding in CNF production Study on several parameters in the production of CNFs with a ball mill	[33] [34] [35]
Cryogenic crushing First, the cellulose pulp is frozen in liquid nitrogen; then, it is crushed	No study on the method itself was found	
Steam explosion The cellulose pulp is introduced into an autoclave at high temperature and subjected to steam for a short period of time; then, pressure discharge occurs	Study on the effect of the steam explosion method on the morphological, chemical, and mechanical properties of CNFs	[37]
Ultrasound Ultrasounds are focused on cellulose pulp and the phenomenon of cavitation (formation, growth, and explosion of bubbles) occurs	Effect of low-frequency ultrasound time on the production of cellulose nanocrystal (CNC) films Influence of the hybrid method of ultrasound and chemical pre-treatment on the dimensions and appearance of CNFs Study of the effect of amplitude and time of ultrasound on the CNCs	[40] [7] [39]
Extrusion Place the cellulose pulp in an extruder with a screw and form a powder	Study of the effect of the number of passages in a double screw extruder on the properties of CNFs Study on the deconstruction of CNFs in an extruder with a double screw with enzymatic hydrolysis in situ (bioextrusion)	[41] [42]
Aqueous counter collision (ACC) Two jets of aqueous cellulose suspension are expelled against each other	Studies of the application of the aqueous collision method (ACC) for the production of CNFs from bacterial cellulose	[46] [43]
Electrospinning Cellulose pulp is ejected into a fine needle between two electrodes. The drops fall and are subject to an electric field	Study of parameters affecting electrospinning Cellulose electrospinning in ionic liquids	[48] [21]
Homogenization, ultrasound, and grinding	Comparative study of homogenization, ultrasound, and grinding processes for CNC preparation	[49]
Microfluidization and grinding	Comparative study of microfluidization and grinding processes for CNC preparation	[50]

**Table 3 materials-16-03104-t003:** Characteristics of CNCs obtained via acid hydrolysis with and without ultrasounds (adapted from [62]).

Method	Features of CNC
Acid hydrolysis—H_2_SO_4_	Size—28 to 470 nm
	Crystallinity index—55.76 ± 7.82%
Acid hydrolysis—H_2_SO_4_ with ultrasound	Size—10 to 50 nm
	Crystallinity index—81.23%

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
