# Peer review of "Nanotechnology Applied to Cellulosic Materials"

_materials, 2023, doi:10.3390/ma16083104_

Round 1
Reviewer 1 Report
The author describes the mechanical, chemical and enzymatic methods needed to produce nanocellulose and their applications. Based on the content of the review, this work is suitable for publication in the materials. However, some major revisions should be addressed before reconsideration. Several suggestions have been given below:
1.In this abstract, what do“TIME, APS, SPS and TENGs”represent respectively? The full names should be given when they first appear in the paper.
2. In the Introduction, the logic of the background introduction is unclear. Therefore, the Introduction requires significant revision to ensure logical clarity and to highlight the novelty and importance of this work.
3. In line 72-73, "APS (ammonium persulfate) and SPS (sodium persulfate)" should be" ammonium persulfate (APS) and sodium persulfate (SPS)". The following similarities are suggested to be revised one by one.
4. In line 73, ", [13–16]" should be "[13–16]". The following similarities are suggested to be revised one by one.
5. In line 289 and 313, "180-210ºC" should be " 180-210 ºC", "5000ºC" should be " 5000 ºC", The following similarities are suggested to be revised one by one.
6. "Recently, researchers Shojaeiarani et al, (2020) have evaluated the effect of the amplitude and time of ultrasounds on the morphology of cellulose nanocrystals and their dispersion. They concluded that the increase in amplitude causes a marked decrease in the length of the CNC, while the increase in the time interval of the ultrasounds slightly affects the length of the CNC". This paragraph needs to be cited with references.
7. In line 507, "researchers [63] [63] aimed to evaluate the effect of different types of pre-treatments on jute fibers" should be " researchers [63] aimed to evaluate the effect of different types of pre-treatments on jute fibers".
8. In line 777, " Zn 2+" should be " Zn2+".
Author Response
The author describes the mechanical, chemical and enzymatic methods needed to produce nanocellulose and their applications. Based on the content of the review, this work is suitable for publication in the materials. However, some major revisions should be addressed before reconsideration. Several suggestions have been given below:
- In this abstract, what do“TIME, APS, SPS and TENGs”represent respectively? The full names should be given when they first appear in the paper.
R: This was done as suggested
- In the Introduction, the logic of the background introduction is unclear. Therefore, the Introduction requires significant revision to ensure logical clarity and to highlight the novelty and importance of this work.
R: The Introduction section has been revised.
- In line 72-73, "APS (ammonium persulfate) and SPS (sodium persulfate)"should be" ammonium persulfate (APS) and sodium persulfate (SPS)". The following similarities are suggested to be revised one by one.
R: This was done as suggested
- In line 73, ", [13–16]" should be "[13–16]". The following similarities are suggested to be revised one by one.
R: This was done as suggested
- In line 289 and 313, "180-210ºC"should be " 180-210 ºC", "5000ºC" should be " 5000 ºC", The following similarities are suggested to be revised one by one.
R: This was done as suggested
- "Recently, researchers Shojaeiarani et al, (2020) have evaluated the effect of the amplitude and time of ultrasounds on the morphology of cellulose nanocrystals and their dispersion. They concluded that the increase in amplitude causes a marked decrease in the length of the CNC, while the increase in the time interval of the ultrasounds slightly affects the length of the CNC". This paragraph needs to be cited with references.
R: This was done as suggested
- In line 507, "researchers [63] [63] aimed to evaluate the effect of different types of pre-treatments on jute fibers" should be " researchers [63] aimed to evaluate the effect of different types of pre-treatments on jute fibers".
R: This was done as suggested
- In line 777, " Zn 2+"should be " Zn2+".
R: This was done as suggested
Reviewer 2 Report
This review focuses on the nanocellulose materials, including the production methods and the applications. However, there are still some issues must be addressed before accepted:
1. The present paper only has 2 figures, lacking readability. More figures should be provided, (eg: typical morphologies of different nanocelluloses, mechanism and performance of nanocellulose applications…)
2. Why TENGs was particular emphasized? Is there any special characteristics of nanocellulose for this application?
3. The referred work of nanocellulose applications was limited of publications in 2021. A fully survey of nanocellulose applications should be provided. More references should be cited.
4. The main challenges and prospects should be summarized in the "conclusion" section.
5. The English language is poor. Too many grammar mistakes exist. Please have a careful check of the full manuscript.
Author Response
This review focuses on the nanocellulose materials, including the production methods and the applications. However, there are still some issues must be addressed before accepted:
- The present paper only has 2 figures, lacking readability. More figures should be provided, (eg: typical morphologies of different nanocelluloses, mechanism and performance of nanocellulose applications…)
R: New figures were added as suggested to increase the readability of the paper.
- Why TENGs was particular emphasized? Is there any special characteristics of nanocellulose for this application?
R: Because everything indicates that the new era will pass through these materials, because it transforms kinetic energy into energy.
- The referred work of nanocellulose applications was limited of publications in 2021. A fully survey of nanocellulose applications should be provided. More references should be cited.
R: More recent references were added as suggested, some of them published in 2023.
- The main challenges and prospects should be summarized in the "conclusion" section.
R: The conclusions were rewritten to respond to the suggestion.
- The English language is poor. Too many grammar mistakes exist. Please have a careful check of the full manuscript.
R: The English has been revised thoroughly.
Reviewer 3 Report
The paper is quite extensive and has certain merits as it provides a sum of information on nanostructured cellulose, but nothing new, given the amount of reports and reviews recently published all over the world, including MDPI, on the preparation and modification of nanocelluloses.
There are also some serious flaws that cannot be accepted for a scientific report, mainly a review:
- very poor English language and style (incomplete sentences - "The first concerns production methods and the second on the applications.", confusing phrases - "This revolution will allow applications that a few years ago would be unthinkable applications.", inaccurate expressions - "...nanocelluloses have attracted research given the versatility of applications, biodegradability, availability, and also biocompatibility" - the term versatility can be used only for a material, not a property!, inappropriate use of terms, such as "expose" instead of "present", etc.); it is easy to discriminate which sections have been written by authors and which ones have been imported from literature;
- serious mistakes have been noticed, such as: TIME-mediated oxidation - in fact, is TEMPO-assisted oxidation; the entire section 2 is full of unclear or incorrect principles and concepts (in example, one paragraph: "As for the cellulose nanofibers (CNF) synonyms used are: cellulose nanofibrils, nanofibrillated cellulose and sometimes even microfibrillated cellulose and microfibrils [2,5,21,22]. Relatively, bacterial cellulose (BC) is also called microbial cellulose or biocellulose [2,14,21,22]." - to mistake nanometric scale for the micrometric one, to use inappropriate terms as biocellulose, and even to use the work of Professor Alain Dufresne to justify some errors, it is not something to tolerate!);
- careless editing: in example, the term TENGs is introduced from the very beginning without being made explicit, but the triboelectric nanogenerators are not discussed at length until section 7.
In my opinion and given the virtues and flaws of this manuscript, the best option is to reject it and ask authors to re-write it in a more serious, focused manner, taking into consideration to significantly reduce the length of the first 6 sections and to expand their literature survey on TENGs.
Author Response
The paper is quite extensive and has certain merits as it provides a sum of information on nanostructured cellulose, but nothing new, given the amount of reports and reviews recently published all over the world, including MDPI, on the preparation and modification of nanocelluloses.
There are also some serious flaws that cannot be accepted for a scientific report, mainly a review:
- very poor English language and style (incomplete sentences - "The first concerns production methods and the second on the applications.", confusing phrases - "This revolution will allow applications that a few years ago would be unthinkable applications.", inaccurate expressions - "...nanocelluloses have attracted research given the versatility of applications, biodegradability, availability, and also biocompatibility" - the term versatility can be used only for a material, not a property!, inappropriate use of terms, such as "expose" instead of "present", etc.); it is easy to discriminate which sections have been written by authors and which ones have been imported from literature;
R: The English has been revised thoroughly and all the mentioned errors have been corrected.
- serious mistakes have been noticed, such as: TIME-mediated oxidation - in fact, is TEMPO-assisted oxidation; the entire section 2 is full of unclear or incorrect principles and concepts (in example, one paragraph: "As for the cellulose nanofibers (CNF) synonyms used are: cellulose nanofibrils, nanofibrillated cellulose and sometimes even microfibrillated cellulose and microfibrils [2,5,21,22]. Relatively, bacterial cellulose (BC) is also called microbial cellulose or biocellulose [2,14,21,22]." - to mistake nanometric scale for the micrometric one, to use inappropriate terms as biocellulose, and even to use the work of Professor Alain Dufresne to justify some errors, it is not something to tolerate!);
R: Some translation mistakes such as tempo which is the Portuguese word for time were corrected. The phrase has been rewritten although there was no confusion about the micro and nanoscale we only tried to reproduce the words of Professor Alain Dufresne “….cellulose nanofibrils (CNF), with the synonyms of nanofibrillated cellulose (NFC), microfibrillated cellulose (MFC), cellulose nanofibers…..”
- careless editing: in example, the term TENGs is introduced from the very beginning without being made explicit, but the triboelectric nanogenerators are not discussed at length until section 7.
In my opinion and given the virtues and flaws of this manuscript, the best option is to reject it and ask authors to re-write it in a more serious, focused manner, taking into consideration to significantly reduce the length of the first 6 sections and to expand their literature survey on TENGs.
R: The literature survey on TENGs was considerably expanded as suggested.
Reviewer 4 Report
This review is devoted to the application of nanotechnologies to cellulosic materials. The subject of this review is quite broad on the one hand and relevant - on the other. In this regard, the authors in the title point to two trending topics - nanotechnology and cellulose materials. In terms of volume and subject matter, the article meets the requirements of the Journal. However, there are some points that it is desirable to improve:
1. The paragraph "3.2. Chemical Methods" does not seem to be complete enough. In particular, the works of Prof. Sudakova to obtain nanocellulose and microfibrillated cellulose by delignification. In addition, there are a number of other large scientific groups working on this topic, and this must also be taken into account.
2. In the introduction, it is desirable to indicate the various uses of cellulose: hydrolysis, hydrogenation, sulfation, oxidation, and other modifications. At this point, it is desirable to quote: 10.1007/s00226-022-01363-4 and others.
3. The use of tables by the authors is commendable. They make it easier to perceive the material and generalizations. I recommend that authors add tables to other main points of the article, for example in 3.2.
4. There are a sufficient number of scientific reviews on the subject of cellulose-based materials. Authors need to be more clear about the benefits of their work.
5. The authors have done a serious job of summarizing and presenting important points in this topic. It is desirable to indicate more clearly the further prospects for the development of nanotechnologies for the production of cellulose and its derivatives.
Author Response
- The paragraph "3.2. Chemical Methods" does not seem to be complete enough. In particular, the works of Prof. Sudakova to obtain nanocellulose and microfibrillated cellulose by delignification. In addition, there are a number of other large scientific groups working on this topic, and this must also be taken into account.
R: The paragraph 3.2 was refined by adding the work of Prof Sudakova and others as suggested
- In the introduction, it is desirable to indicate the various uses of cellulose: hydrolysis, hydrogenation, sulfation, oxidation, and other modifications. At this point, it is desirable to quote: 10.1007/s00226-022-01363-4 and others.
R: The various uses of cellulose: hydrolysis, hydrogenation, sulfation, oxidation, and other modifications were included citing the mentioned article, and others, not in the introduction, since there are too many, but in section 6 “Various applications of nanocellulose”.
- The use of tables by the authors is commendable. They make it easier to perceive the material and generalizations. I recommend that authors add tables to other main points of the article, for example in 3.2.
R: The reason why no more tables were added is that the article has already 39 pages and additional tables would increase the length of the paper that was already reduced by suggestion of another reviewer.
- There are a sufficient number of scientific reviews on the subject of cellulose-based materials. Authors need to be more clear about the benefits of their work.
R: This review is one of the most complete reviews dealing with the methods for obtaining nanocellulose namely: cellulose nanocrystals (CNC), cellulose nanofibers (CNF) and bacterial cellulose (BC) and their preparation methods with insights from several very recent articles (2023) not discussed in similar publications. This was better emphasized in the conclusion section.
- The authors have done a serious job of summarizing and presenting important points in this topic. It is desirable to indicate more clearly the further prospects for the development of nanotechnologies for the production of cellulose and its derivatives.
R: The further prospects for the development of nanotechnologies for the production of cellulose and its derivatives were summarized in the section Challenges and opportunities, after the Conclusions section.
Reviewer 5 Report
The paper is about the nano-sized cellulose form, where author extensively described the process methods. There are few minor comments that would improve the paper.
1. Please improve the quality of the figure-1, Figure-2 and Figure 3
2. It seems the paper was thoroughly revised and looks refined; however, there are still some typographical errors in the manuscript.
3. One of the points that I found that the authors need to consider when talking about changes in the physicochemical properties is how the reactivity of cellulose in different forms alters when processed through different methods. A small section or table would be sufficient and help readers to engage with the manuscript.
4. Bacterial cellulose has distinctive characteristics than plant-derived cellulose. Paper would be more appealing if author write a small section comparing bacterial cellulose reactivity with plant-derived cellulose.
The paper is in good shape and can be considered after author revised these minor changes.
Author Response
First of all the authors would like to thank for all the comments.
- Please improve the quality of the figure-1, Figure-2 and Figure 3
R: The figures are already in photographic quality. If needed, the size of the figures can be increased in order to make them more visible.
- It seems the paper was thoroughly revised and looks refined; however, there are still some typographical errors in the manuscript.
R: The manuscript has been revised for typographical errors as suggested.
- One of the points that I found that the authors need to consider when talking about changes in the physicochemical properties is how the reactivity of cellulose in different forms alters when processed through different methods. A small section or table would be sufficient and help readers to engage with the manuscript.
R It is worth noting that the effect of preparation method on the structural features and some physicochemical properties of obtained CNF and CNC is already carefully reviewed in the section 3 (Nanocelluloses preparation methods), including the summary of some basic properties (Table 5). The effect of preparation method on the reactivity of cellulose in particular chemical reactions is out of the scope of this review, as it is pointed out in the introductory part of the section 4 (where the specific review on this matter are recommended).
- Bacterial cellulose has distinctive characteristics than plant-derived cellulose. Paper would be more appealing if author write a small section comparing bacterial cellulose reactivity with plant-derived cellulose.
R: It is true that BC have differ characteristics than plant-derived cellulose. These features were highlighted in the manuscript (e.g., in the section 3). The effect of BC on the properties of materials produced thereof, in comparison to plant-derived cellulose preparations, are reviewed in the sections 5 and 6 within the scope of specific applications thus reflecting structure-properties relationships. The reactivity of BC in comparison to plant-derived cellulose in particular chemical reactions is out of the scope of this review.
The paper is in good shape and can be considered after author revised these minor changes.
Round 2
Reviewer 2 Report
This manuscript focused on the unique properties, application of nanocelluloses has been viewed. Various types of chemical functionalization are referred to as a strategy for modifying the properties of nanocelluloses. It is suitable for published.
Author Response
Thank you for your kind comments.